# Nearly-Tight and Oblivious Algorithms for Explainable Clustering

**Buddhima Gamlath**[*]
EPFL
buddhima.gamlath@epfl.ch

**Xinrui Jia**[*]
EPFL
xinrui.jia@epfl.ch

**Adam Polak**[*]
EPFL
adam.polak@epfl.ch

**Ola Svensson**[*]
EPFL
ola.svensson@epfl.ch

## Abstract

We study the problem of explainable clustering in the setting first formalized by Dasgupta, Frost, Moshkovitz, and Rashtchian (ICML 2020). A $k$-clustering is said to be explainable if it is given by a decision tree where each internal node splits data points with a threshold cut in a single dimension (feature), and each of the $k$ leaves corresponds to a cluster. We give an algorithm that outputs an explainable clustering that loses at most a factor of $O(\log^2 k)$ compared to an optimal (not necessarily explainable) clustering for the $k$-medians objective, and a factor of $O(k \log^2 k)$ for the $k$-means objective. This improves over the previous best upper bounds of $O(k)$ and $O(k^2)$, respectively, and nearly matches the previous $\Omega(\log k)$ lower bound for $k$-medians and our new $\Omega(k)$ lower bound for $k$-means. The algorithm is remarkably simple. In particular, given an initial not necessarily explainable clustering in $\mathbb{R}^d$, it is oblivious to the data points and runs in time $O(dk \log^2 k)$, independent of the number of data points $n$. Our upper and lower bounds also generalize to objectives given by higher $\ell_p$-norms.

## 1 Introduction

An important topic in current machine learning research is understanding how models actually make their decisions. For a recent overview on the subject of explainability and interpretability, see, e.g., [13, 14]. Many good methods exist (e.g. [15]) for interpreting black-box models, so called *post-modeling explainability*, but this approach has been criticized [16] for providing little insight into the data. Currently, there is a shift towards designing models that are interpretable by design.

Clustering is a fundamental problem in unsupervised learning. A common approach to clustering is to minimize the $k$-medians or $k$-means objectives, e.g., with the celebrated Lloyd's [11] or $k$-means++ [2] algorithms. Both objectives are also widely studied from a theoretical perspective, and, in particular, they admit constant-factor approximation algorithms running in polynomial time [1, 3, 4, 9].

In their recent paper [6], Dasgupta et al. were the first to study provable guarantees for explainable clustering. They define a $k$-clustering to be *explainable* if it is given by a decision tree, where each internal node splits data points with a *threshold cut* in a single dimension (feature), and each of the $k$ leaves corresponds to a cluster (see Figure 1).

---

[*]Equal contribution

35th Conference on Neural Information Processing Systems (NeurIPS 2021).

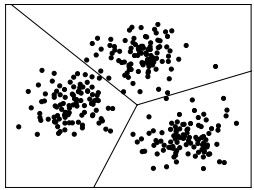
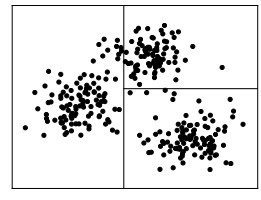
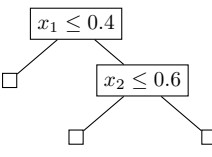

(a) Non-explainable clustering      (b) Explainable clustering      (c) Threshold tree

Figure 1: Examples of an optimal non-explainable and a costlier explainable clustering of the same set of points in $\mathbb{R}^2$, together with the threshold tree defining the explainable clustering.

This definition is motivated by the desire to have a concise and easy-to-explain reasoning behind how the model chooses data points that form a cluster. See the original paper [6] for an extensive discussion of motivations and a survey of previous (empirical) approaches to explainable clustering.

The central question to study in this setting is that of the *price of explainability*: How much do we have to lose – in terms of a given objective, e.g., $k$-medians or $k$-means – compared to an optimal unconstrained clustering, if we insist on an explainable clustering, and can we efficiently construct such a clustering?

Dasgupta et al. [6] proposed an algorithm that, given an unconstrained (non-explainable) *reference clustering*[2], produces an explainable clustering losing at most a multiplicative factor of $O(k)$ for the $k$-medians objective and $O(k^2)$ for $k$-means, compared to the reference clustering. They also gave a lower bound showing that an $\Omega(\log k)$ loss is unavoidable, both for the $k$-medians and $k$-means objective. Later, Laber and Murtinho [10] improved over the upper bounds in a low-dimensional regime $d \leq k/\log(k)$, giving an $O(d \log k)$-approximation algorithm for explainable $k$-medians and an $O(dk \log k)$-approximation algorithm for explainable $k$-means.

## 1.1 Our contributions

**Improved clustering cost.** We present a randomized algorithm that, given $k$ centers defining a reference clustering and a number $p \geq 1$, constructs a threshold tree that defines an explainable clustering that is, in expectation, worse than the reference clustering by at most a factor of $O(k^{p-1} \log^2 k)$ for the objective given by the $\ell_p$-norm. That is $O(\log^2 k)$ for $k$-medians and $O(k \log^2 k)$ for $k$-means.

**Simple and oblivious algorithm.** Our algorithm is remarkably simple. It samples threshold cuts uniformly at random (for $k$-medians; $k$-means and higher $\ell_p$-norms need slightly fancier distributions) until all centers are separated from each other. In particular, the input to the algorithm includes only the centers of a reference clustering and not the data points.

As a consequence, the algorithm cannot overfit the data (any more than the reference clustering possibly already does), and the same expected cost guarantees hold for any future data points not known at the time of the clustering construction. Besides, the algorithm is fast; its running time does not depend on the number of data points $n$. A naive implementation runs in time $O(dk^2)$, and in Section 3.2, we show how to improve it to $O(dk \log^2 k)$ time, which is near-linear in the input size $dk$ of the $k$ reference centers.

**Nearly-tight bounds.** We complement our results with a lower bound. We show how to construct instances of the clustering problem such that any explainable clustering must be at least $\Omega(k^{p-1})$ times worse than an optimal clustering for the $\ell_p$-norm objective. In particular, this improves the previous $\Omega(\log k)$ lower bound for $k$-means [6] to $\Omega(k)$ .

---

[2]A reference clustering can be obtained, e.g., by running a constant-factor approximation algorithm for a given objective function. Then, the asymptotic upper bounds of the explainable clustering cost compared to the reference clustering translate identically to the bounds when compared to an optimal clustering.

Table 1: **Algorithms and lower bounds for explainable $k$-clustering in $\mathbb{R}^d$.** For a given objective function, how large a multiplicative factor do we have to lose, compared to an optimal unconstrained clustering, if we insist on an explainable clustering?

| | $k$-medians | $k$-means | $\ell_p$-norm | |
|---|---|---|---|---|
| **Algorithms** | $O(k)$ | $O(k^2)$ | | Dasgupta et al. [6] |
| | $O(d \log k)$ | $O(kd \log k)$ | | Laber and Murtinho [10] |
| | $O(\log^2 k)$ | $O(k \log^2 k)$ | $O(k^{p-1} \log^2 k)$ | **This paper** |
| | $O(\log k \log \log k)$ | $O(k \log k \log \log k)$ | | Makarychev and Shan [12] |
| | $O(\log k \log \log k)$ | $O(k \log k)$ | | Esfandiari et al. [7] |
| | $O(d \log^2 d)$ | | | Esfandiari et al. [7] |
| | | $O(k^{1-2/d} \operatorname{polylog} k)$ | | Charikar and Hu [5] |
| **Lower bounds** | $\Omega(\log k)$ | $\Omega(\log k)$ | | Dasgupta et al. [6] |
| | | $\Omega(k)$ | $\Omega(k^{p-1})$ | **This paper** |
| | | $\Omega(k/\log k)$ | | Makarychev and Shan [12] |
| | $\Omega(\min(d, \log k))$ | $\Omega(k)$ | | Esfandiari et al. [7] |
| | | $\Omega(k^{1-2/d}/\operatorname{polylog} k)$ | | Charikar and Hu [5] |

In consequence, we give a nearly-tight answer to the question of the price of explainability. We leave a $\log(k)$ gap for $k$-medians, and a $\log^2(k)$ gap for $k$-means and higher $\ell_p$-norm objectives. See Table 1 for a summary of the upper and lower bounds discussed above and recent independent works discussed in Section 1.3.

## 1.2 Technical overview

The theoretical guarantees obtained by Dasgupta et al. [6] depend on the number of clusters $k$ and the height of the threshold tree obtained $H$. Their algorithm loses, compared to the input reference clustering, an $O(H)$ factor for the $k$-medians cost and $O(Hk)$ for $k$-means. These approximations are achieved by selecting a threshold cut that separates some two centers and minimizes the number of points that get separated from their centers in the reference clustering. This creates two children of a tree node, and the threshold tree is created by recursing on each of the children. The height of the tree $H$ may need to be $k - 1$. For example, consider the data set in $\mathbb{R}^k$ consisting of the $k$ standard basis vectors (see Figure 2). Laber and Murtinho [10] replace the dependence on $H$ with $d$, the dimension of the data set, by first constructing optimal search trees for each dimension and then carefully using them to guide the construction of the threshold tree. In our work, we obtain improved guarantees by using randomized cuts that are oblivious to the data points and depend only on the reference centers, in contrast to the above-mentioned two prior approaches, which selected cuts based on the data points.

There are two components to achieving our improved guarantees that correspond with two aspects of the minimum cut algorithm of [6]: the use of the minimum cut, and the height of the threshold tree produced. The first observation is that, for the $\ell_1$-norm, we do not lose in the analysis by taking a cut uniformly at random compared to always using the minimum cut. (The corresponding distribution for higher $\ell_p$-norms is proportional to the $p$-th power of the distance to the closest center.) Indeed, using a random cut makes us robust against specifically engineered examples, such as the one that fools the minimum cut algorithm of [6] (see Appendix A in the supplementary material). In that example we add dimensions in which a cut is mini-

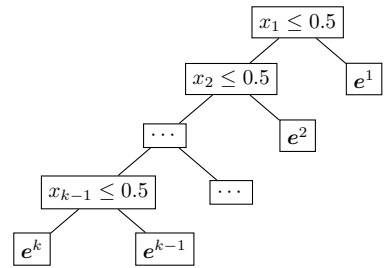

Figure 2: An optimal threshold tree for the $k$ standard basis vectors in $\mathbb{R}^k$. Any optimal threshold tree on this data set has height $k - 1$.

mum, but these minimum cuts produce a tree of height $\Omega(k)$ whose cost is $\Omega(k)$ times larger than the optimum.

However, threshold trees of height $\Omega(k)$ are unavoidable in certain instances as seen in the example with $k$ standard basis vectors (Figure 2). This leads to our second observation that it is necessary to use a tighter upper bound on the cost of reassigning a point since any height $k-1$ threshold tree produced on this example is actually optimal. Using the diameter definition in [6], the cost of each cut is upper bounded by $k$ while the actual distance between any two centers is at most 2, which is also a valid upper bound for the reassignment cost. Hence, we use the maximum distance between any two centers to upper bound the cost of misclassifying a point.

**Limitations and further work.** We conjecture that our $k$-medians algorithm is asymptotically optimal. In particular, we believe the actual approximation ratio of our algorithm is $1 + H_{k-1}$, where $H_n$ is the $n$-th harmonic number (recall that $\ln(n) \leq H_n \leq 1 + \ln(n)$). There are two potential barriers in our current analysis that prevent us from demonstrating this optimality. The first is that our upper bound on the cost increase of assigning a single point to a wrong center is not tight, and secondly, our analysis may include the cost of the same point multiple times. Despite the further developments mentioned in Section 1.3, it still remains to fully resolve the correct asymptotic price of explainability.

Some potential directions to expanding our work include parallelizations, generalizing the notion of explainability, and defining natural clusterability assumptions under which the price of explainability is reduced. Constructing a threshold tree seems inherently sequential; it would be interesting to explore parallelizations for faster implementation. Another direction would be to allow each node to be a hyperplane in a chosen number of dimensions instead of only splitting along one feature. Finally, it seems a non-trivial question to find a right clusterability assumption on the data points distribution – that would allow us to overcome the existing lower bounds – because these lower bounds are in fact very "clusterable" instances, in the traditional usage of this notion.

**Societal impact.** Our contributions are theoretical in nature and thus are not tailored to any specific application. We show that a simple algorithm has good approximation guarantees while also having a running time independent of the number of points in the data set. Moreover, this algorithm is oblivious to the data points, which avoids introducing biases. However, our algorithms rely on a reference clustering and may show existing biases of the original clustering.

## 1.3 Independent work

We note independent further developments by Makarychev and Shan [12]; Esfandiari, Mirrokni, and Narayanan [7]; and Charikar and Hu [5].

Makarychev and Shan [12] showed $O(\log k \log \log k)$ and $O(k \log k \log \log k)$ upper bounds for $k$-medians and $k$-means, respectively, thus improving over our bounds by a factor of $\log k / \log \log k$. Their $k$-medians algorithm is essentially the same as our *modified* Algorithm 1 (see Section 3.1.1), but they provide a tighter analysis. Their $k$-means upper bound follows from combining their $k$-medians algorithm with their insightful reduction from $k$-means to $k$-medians that loses a factor of $O(k)$. However, the $k$-means algorithm resulting from that combination is essentially the same as our Algorithm 2. They also provide an $\Omega(k / \log k)$ lower bound for $k$-means, which is slightly worse than ours. Finally, they study the explainable $k$-medoids problem (i.e., $k$-medians with $\ell_2$ norm), and provide an $O(\log^{3/2} k)$ upper bound and an $\Omega(\log k)$ lower bound.

Esfandiari, Mirrokni, and Narayanan [7] also give an $O(\log k \log \log k)$ upper bound for $k$-medians. Their algorithm is essentially the same as our (unmodified) Algorithm 1, and, again, they provide a tighter analysis. They also give an $O(k \log k)$ upper bound for $k$-means, improving over the result of Makarychev and Shan by a factor of $\log \log k$. Their $k$-means algorithm is similar to our Algorithm 2 but samples cuts from a different distribution. They also match our $\Omega(k)$ lower bound for $k$-means, and improve the $k$-medians lower bound of Dasgupta et al. [6] to $\Omega(\min(d, \log k))$.

Charikar and Hu [5] focus on explainable $k$-means and present an $O(k^{1-2/d} \operatorname{poly}(d \log k))$-approximation algorithm, which is better than any previous algorithm when $d = O(\log k / \log \log k)$ (in particular, for any constant dimension $d$). Resorting to an $O(k \operatorname{polylog} k)$-approximation algorithm

(e.g., [12]) when this is not the case, they obtain an $O(k^{1-2/d} \operatorname{polylog} k)$ upper bound. They match it with an $\Omega(k^{1-2/d}/\operatorname{polylog} k)$ lower bound, which is tight up to polylogarithmic factors.

## 2   Preliminaries

Following the notation of [6], we use bold variables for vector values and corresponding non-bold indexed variables for scalar coordinates. Intuitively, a clustering is explainable because the inclusion of a data point $\boldsymbol{x} = [x_1, \ldots, x_d]$ to a particular cluster is "easily explained" by whether or not $\boldsymbol{x}$ satisfies a series of inequalities of the form $x_i \leq \theta$. These inequalities are called **threshold cuts**, defined by a coordinate $i \in [d]$ (denoting the set $\{1, 2, \ldots, d\}$) and a threshold $\theta \in \mathbb{R}$. More precisely, a **threshold tree** is a binary tree where each non-leaf node is a threshold cut $(i, \theta)$ which assigns the point $\boldsymbol{x}$ of that node into the left child if $x_i \leq \theta$ and the right child otherwise. A clustering is **explainable** if the clusters are in bijection to the leaves of a threshold tree with exactly $k$ leaves that started with all the data points at the root.

Given a set of points $\mathcal{X} = \{\boldsymbol{x}^1, \boldsymbol{x}^2, \ldots, \boldsymbol{x}^n\} \subseteq \mathbb{R}^d$ and its clustering $\{C^1, \ldots, C^k\}, \bigcup_{j=1}^k C^j = \mathcal{X}$, the $k$-**medians cost** of the clustering is defined in [6] as

$$\operatorname{cost}_1(C^1, \ldots, C^k) = \sum_{j=1}^k \min_{\boldsymbol{\mu} \in \mathbb{R}^d} \sum_{\boldsymbol{x} \in C^j} \|\boldsymbol{x} - \boldsymbol{\mu}\|_1 = \sum_{j=1}^k \sum_{\boldsymbol{x} \in C^j} \|\boldsymbol{x} - \operatorname{median}(C^j)\|_1.$$

The $k$-**means** cost is defined analogously with the square of the $\ell_2$ distance of every point to $\operatorname{mean}(C^j)$.

For a set of centers $\mathcal{U} = \{\boldsymbol{\mu}^1, \ldots, \boldsymbol{\mu}^k\} \subseteq \mathbb{R}^d$, a non-explainable clustering $\{\widetilde{C}^1, \ldots, \widetilde{C}^k\}$ of $\mathcal{X}$ is given by $\widetilde{C}^j = \{\boldsymbol{x} \in \mathcal{X} \mid \boldsymbol{\mu}^j = \arg\min_{\boldsymbol{\mu} \in \mathcal{U}} \|\boldsymbol{x} - \boldsymbol{\mu}\|_1\}$, and we write $\operatorname{cost}_1(\mathcal{U}) = \operatorname{cost}_1(\widetilde{C}^1, \ldots, \widetilde{C}^k)$. Note that $\operatorname{cost}_1(\mathcal{U}) = \sum_{\boldsymbol{x} \in \mathcal{X}} \min_{\boldsymbol{\mu} \in \mathcal{U}} \|\boldsymbol{x} - \boldsymbol{\mu}\|_1$.

Given a threshold tree $T$, the leaves of $T$ induce an explainable clustering $\{\widehat{C}^1, \ldots, \widehat{C}^k\}$, and we write $\operatorname{cost}_1(T) = \operatorname{cost}_1(\widehat{C}^1, \ldots, \widehat{C}^k)$. In the analyses, however, we often upper bound the cost of each explainable cluster $\widehat{C}^j$ using the corresponding reference center $\boldsymbol{\mu}^j$:

$$\operatorname{cost}_1(T) \leq \sum_{j=1}^k \sum_{\boldsymbol{x} \in \widehat{C}^j} \|\boldsymbol{x} - \boldsymbol{\mu}^j\|_1.$$

These may not be optimal center locations, yet we are still able to obtain guarantees that are polylog away from being tight.

We generalize the above to higher $\ell_p$-norms, $p \geq 1$, as follows

$$\operatorname{cost}_p(C^1, \ldots, C^k) = \sum_{j=1}^k \min_{\boldsymbol{\mu} \in \mathbb{R}^d} \sum_{\boldsymbol{x} \in C^j} \|\boldsymbol{x} - \boldsymbol{\mu}\|_p^p, \qquad \operatorname{cost}_p(T) \leq \sum_{j=1}^k \sum_{\boldsymbol{x} \in \widehat{C}^j} \|\boldsymbol{x} - \boldsymbol{\mu}^j\|_p^p.$$

## 3   Explainable $k$-medians clustering

In this section we present our algorithm for explainable $k$-medians and its analysis. Recall that our algorithm is oblivious to the data points: It determines the threshold tree using only the center locations. The algorithm simply samples a sequence of cuts until it defines a threshold tree with each center belonging to exactly one leaf. In what follows, we elaborate on this process in detail.

The algorithm's input is a set of centers $\mathcal{U} = \{\boldsymbol{\mu}^1, \boldsymbol{\mu}^2, \ldots, \boldsymbol{\mu}^k\} \subset \mathbb{R}^d$. We consider cuts that intersect the bounding box of $\mathcal{U}$. Letting $\mathrm{I}_i = [\min_{j \in [k]} \mu_i^j, \max_{j \in [k]} \mu_i^j]$ be the interval between the minimum and maximum $i$-coordinate of centers, the set of all possible cuts that intersect the bounding box of $\mathcal{U}$ is $\operatorname{AllCuts} = \{(i, \theta) : i \in [d], \theta \in \mathrm{I}_i\}$. Our algorithm uses a stream of independent uniformly random cuts from AllCuts. In particular, the probability density function of $(i, \theta) \in \operatorname{AllCuts}$ is $1/L$ where $L = \sum_{i \in [d]} |I_i|$ is the sum of the side lengths of the bounding box of $\mathcal{U}$.

The algorithm simply takes cuts from this stream until it produces a threshold tree. To this end, it maintains a tentative set of tree leaves, each identified by a subset of centers, and continues until it has

$k$ leaves of singleton sets. We say a cut *splits* a leaf if the cut properly intersects with the bounding box of the corresponding subset of centers. In other words, a cut $(i, \theta)$ splits a leaf $B$ if and only if the two sets $B^- = \{\boldsymbol{\mu} \in B : \mu_i \leq \theta\}$ and $B^+ = \{\boldsymbol{\mu} \in B : \mu_i > \theta\}$ are both non-empty. At the beginning, the algorithm starts with a single leaf identified by $\mathcal{U}$, the set of all centers. It then samples a cut $(i, \theta)$ and checks if it splits any existing leaf. If so, it saves the cut, and for each leaf $B$ that gets split by the cut into $B^-$ and $B^+$, adds $B^-$ and $B^+$ as two new leaves rooted at $B$. These saved cuts define the output threshold tree. We present the pseudo-code of this algorithm in Algorithm 1.

---

**Algorithm 1:** Explainable $k$-medians algorithm.

1 **Input:** A collection of $k$ centers $\mathcal{U} = \{\boldsymbol{\mu}^1, \boldsymbol{\mu}^2, \ldots, \boldsymbol{\mu}^k\} \subset \mathbb{R}^d$.
2 **Output:** A threshold tree with $k$ leaves.
3 Leaves $\leftarrow \{\mathcal{U}\}$
4 **while** $|\text{Leaves}| < k$ **do**
5      Sample $(i, \theta)$ uniformly at random from AllCuts.
6      **for** *each* $B \in$ Leaves *that are split by* $(i, \theta)$ **do**
7          Split $B$ into $B^-$ and $B^+$ and add them as left and right children of $B$.
8          Update Leaves.

9 **return** the threshold tree defined by all cuts that separated some $B$.

---

### 3.1 Cost analysis

We show that Algorithm 1 satisfies the following guarantees.

**Theorem 1.** *Given reference centers $\mathcal{U} = \{\boldsymbol{\mu}^1, \boldsymbol{\mu}^2, \ldots, \boldsymbol{\mu}^k\}$, Algorithm 1 outputs a threshold tree $T$ whose expected cost satisfies*

$$\mathbb{E}[\text{cost}_1(T)] \leq O(\log(k) \cdot (1 + \log(c_{\max}/c_{\min}))) \cdot \text{cost}_1(\mathcal{U}),$$

*where $c_{\max}$ and $c_{\min}$ denote the maximum and minimum pairwise distance between two centers in $\mathcal{U}$, respectively. Furthermore, with probability at least $1 - 1/k$, Algorithm 1 outputs a threshold tree $T$ from a distribution with $\mathbb{E}[\text{cost}_1(T)] \leq O(\log^2 k) \cdot \text{cost}_1(\mathcal{U})$.*

The (more involved) proof of the furthermore statement is discussed in Section 3.1.1 and formally proved in Appendix B in the supplementary material. We remark that the success probability $1 - 1/k$ can be made larger by only slightly increasing the hidden constant in the cost guarantee. Furthermore, in Section 3.1.1, we give a slight adaptation of the above algorithm that has an expected cost bounded by $O(\log^2 k) \cdot \text{cost}_1(\mathcal{U})$. The remaining part of this section is devoted to proving the upper bound on the expected cost of Algorithm 1.

**Proof outline.** First, in Lemma 1, we show that a random cut in expectation separates $\text{cost}_1(\mathcal{U})/L$ points from their closest centers. Indeed, note that the probability of separating a point $\boldsymbol{x}$ from its center $\pi(\boldsymbol{x})$ is at most $\|\boldsymbol{x} - \pi(\boldsymbol{x})\|_1/L$, and on the other hand, $\text{cost}_1(\mathcal{U}) = \sum_{x \in \mathcal{X}} \|\boldsymbol{x} - \pi(\boldsymbol{x})\|_1$, hence the bound follows from linearity of expectation. Each such separated point incurs a cost of at most $c_{\max}$. Next, in Lemma 2, we show that with good probability $O(\log(k) \cdot L/c_{\max})$ random cuts separate all pairs of centers that are at distance at least $c_{\max}/2$ from each other. Morally, the cost of halving $c_{\max}$, which we will need to perform $1 + \log(c_{\max}/c_{\min})$ many times, is therefore $\text{cost}_1(\mathcal{U})/L \cdot c_{\max} \cdot O(\log(k) \cdot L/c_{\max}) = O(\log(k)) \cdot \text{cost}_1(\mathcal{U})$, and the bound follows (see Lemma 3).

**Formal analysis of the expected cost.** We first bound the number of points that are separated from their closest center by a random cut. This quantity is important as it upper bounds the number of points whose cost is increased in the final tree due to the considered cut. Recall that $L = \sum_{i=1}^{d} |I_i|$ denotes the total side lengths of the bounding box of the input centers $\mathcal{U}$. We also let $f_i(\theta)$ be the number of points separated from their closest center by the cut $(i, \theta)$.

**Lemma 1.** *We have $\mathbb{E}_{(i, \theta)}[f_i(\theta)] \leq \text{cost}_1(\mathcal{U})/L$ where the expectation is over a uniformly random threshold cut $(i, \theta) \in$ AllCuts.*

*Proof.* For a point $\boldsymbol{x} \in \mathcal{X}$ let $\pi(\boldsymbol{x})$ denote the closest center in $\mathcal{U}$. Then by definition,

$$\mathrm{cost}_1(\mathcal{U}) = \sum_{x \in \mathcal{X}} \|\boldsymbol{x} - \pi(\boldsymbol{x})\|_1 = \sum_{i=1}^{d} \sum_{\boldsymbol{x} \in X} |x_i - \pi(\boldsymbol{x})_i| \,.$$

Moreover, if we let $f_i(\theta)$ be the number of points separated from their closest center by the cut $(i, \theta)$, we can rewrite the cost of a fixed dimension $i$ as follows:

$$\sum_{\boldsymbol{x} \in X} |x_i - \pi(\boldsymbol{x})_i| = \sum_{\boldsymbol{x} \in X} \int_{-\infty}^{\infty} \mathbb{1}[\theta \text{ between } x_i \text{ and } \pi(\boldsymbol{x})_i] d\theta = \int_{-\infty}^{\infty} f_i(\theta) d\theta \,.$$

We thus have $\mathrm{cost}_1(\mathcal{U}) = \sum_{i=1}^{d} \int_{-\infty}^{\infty} f_i(\theta) d\theta$.

At the same time, if we let $[a_i, b_i]$ denote the interval $I_i$, then $\frac{1}{|I_i|} \int_{a_i}^{b_i} f_i(\theta) d\theta$ equals the number of points separated from their closest center by a uniformly random threshold cut $(i, \theta) : \theta \in I_i$ along dimension $i$. Thus the expected number of points separated from their closest center by a uniformly random threshold cut in AllCuts is

$$\sum_{i=1}^{d} \frac{|I_i|}{L} \cdot \frac{1}{|I_i|} \int_{a_i}^{b_i} f_i(\theta) d\theta = \frac{1}{L} \sum_{i=1}^{d} \int_{a_i}^{b_i} f_i(\theta) d\theta \leq \frac{1}{L} \sum_{i=1}^{d} \int_{-\infty}^{\infty} f_i(\theta) d\theta = \mathrm{cost}_1(\mathcal{U})/L \,,$$

where we used $f_i(\theta) \geq 0$ for the inequality. $\qquad\square$

The above lemma upper bounds the expected number of points whose cost increases from a uniformly random threshold cut. We proceed to analyze how much this increase is, in expectation. Let $\mathrm{Leaves}(t)$ denote the state of Leaves at the beginning of the $t$-th iteration of the while loop of Algorithm 1 and let $c_{\max}(t) = \max_{B \in \mathrm{Leaves}(t)} \max_{\boldsymbol{\mu}^i, \boldsymbol{\mu}^j \in B} \|\boldsymbol{\mu}^i - \boldsymbol{\mu}^j\|_1$ denote the maximum distance between two centers that belong to the same leaf at the beginning of the $t$-th iteration. With this notation we have that $\mathrm{Leaves}(1) = \{\mathcal{U}\}$ and that $c_{\max}(1)$ equals the $c_{\max}$ in the statement of Theorem 1. Observe that $c_{\max}(t) \geq c_{\max}(t+1)$ and $c_{\max}(t) = 0$ if $|\mathrm{Leaves}| = k$ (i.e., when each leaf contains exactly one center). Understanding the rate at which $c_{\max}(t)$ decreases is crucial for our analysis because of the following observation: Consider a leaf $B \in \mathrm{Leaves}(t)$ and a point $\boldsymbol{x} \in \mathcal{X}$ that has not yet been separated from its closest center $\pi(\boldsymbol{x}) \in B$. If the threshold cut selected in the $t$-th iteration separates $\boldsymbol{x}$ from $\pi(\boldsymbol{x})$ then the cost of $\boldsymbol{x}$ in the final threshold tree is upper bounded by $\max_{\boldsymbol{\mu} \in B} \|\boldsymbol{x} - \boldsymbol{\mu}\|_1$, which, by the triangle inequality, is at most

$$\max_{\boldsymbol{\mu} \in B} \|\boldsymbol{x} - \pi(\boldsymbol{x})\|_1 + \|\pi(\boldsymbol{x}) - \mu\|_1 \leq \|\boldsymbol{x} - \pi(\boldsymbol{x})\|_1 + c_{\max}(t) \,. \tag{1}$$

In other words, a point that is first separated from its closest center by the threshold cut selected in the $t$-th iteration has a cost increase of at most $c_{\max}(t)$.

**Lemma 2.** *Fix the the threshold cuts selected by Algorithm 1 during the first $t-1$ iterations (this determines the random variable* $\mathrm{Leaves}(t)$ *and thus* $c_{\max}(t)$*). Let* $M = 3\ln(k) \cdot 2L/c_{\max}(t)$*. Then*

$$\Pr[c_{\max}(t+M) \leq c_{\max}(t)/2] \geq 1 - 1/k \,,$$

*where the probability is over the random cuts selected in iterations* $t, t+1, \ldots, t+M-1$.

*Proof.* Consider two centers $\boldsymbol{\mu}^i$ and $\boldsymbol{\mu}^j$ that belong to the same leaf in $\mathrm{Leaves}(t)$. The probability that a uniformly random threshold cut from AllCuts separates these two centers equals $\|\boldsymbol{\mu}^i - \boldsymbol{\mu}^j\|_1/L$. Thus if the centers are at distance at least $c_{\max}(t)/2$, the probability that they are *not* separated by any of $M$ independently chosen cuts is at most

$$\left(1 - \frac{c_{\max}(t)/2}{L}\right)^M = \left(1 - \frac{c_{\max}(t)}{2L}\right)^{3\ln(k) \cdot 2L/c_{\max}(t)} \leq (1/e)^{3\ln(k)} = 1/k^3 \,.$$

There are at most $\binom{k}{2}$ pairs of centers in the leaves of $\mathrm{Leaves}(t)$ at distance at least $c_{\max}(t)/2$. By the union bound, we thus have, with probability at least $1 - 1/k$, that each of these pairs are separated by at least one of the cuts selected in iterations $t, t+1, \ldots, t+M-1$. In that case, any two centers in the same leaf of $\mathrm{Leaves}(t+M)$ are at distance at most $c_{\max}(t)/2$ and so $c_{\max}(t+M) \leq c_{\max}(t)/2$. $\qquad\square$

Equipped with the above lemmas we are ready to analyze the expected cost of the tree output by Algorithm 1. Let $(i_t, \theta_t)$ denote the cut selected by Algorithm 1 in the $t$-th iteration. As argued above in (1), $c_{\max}(t)$ upper bounds the cost increase of the points first separated from their closest center by the $t$-th threshold cut. Hence,

$$\mathbb{E}\left[\mathrm{cost}_1(T)\right] \le \mathrm{cost}_1(\mathcal{U}) + \mathbb{E}\left[\sum_t c_{\max}(t) f_{i_t}(\theta_t)\right],$$

where the sum is over the iterations of Algorithm 1 (and recall that $f_i(\theta)$ denotes the number of points separated from their closest center by the cut $(i, \theta)$). We remark that the right-hand side is an upper bound (and not an exact formula of the cost) for two reasons: first, not every separated point may experience a cost increase of $c_{\max}(t)$, and second, the right-hand side adds a cost increase every time a cut separates a point from its closest center and not only the first time. Nevertheless, we show that this upper bound yields the stated guarantee. We do so by analyzing the expected cost increase of the cuts until $c_{\max}(t)$ has halved. Specifically, let

$$\mathrm{cost\text{-}increase}(r) = \sum_{t\,:\,c_{\max}(t)\in(c_{\max}/2^{r+1},\,c_{\max}/2^r]} c_{\max}(t) f_{i_t}(\theta_t)$$

be the random variable that upper bounds the cost increase caused by the cuts selected during the iterations $t$ when $c_{\max}/2^{r+1} < c_{\max}(t) \le c_{\max}/2^r$. Then

$$\mathbb{E}[\mathrm{cost}_1(T)] \le \mathrm{cost}_1(\mathcal{U}) + \sum_r \mathbb{E}[\mathrm{cost\text{-}increase}(r)],$$

where the sum is over $r$ from 0 to $1 + \lfloor \log_2(c_{\max}/c_{\min}) \rfloor$. The bound on the expected cost therefore follows from the following lemma.

**Lemma 3.** *For every $r$, $\mathbb{E}[\mathrm{cost\text{-}increase}(r)] \le 12\ln(k) \cdot \mathrm{cost}_1(\mathcal{U})$.*

Let $M = 3\ln(k) \cdot 2L/c_{\max}(t)$ as in Lemma 2. Using Lemma 1, one can upper bound the expected cost of $M$ uniformly random cuts in iterations $t, t+1, \ldots, t+M-1$ by $6\ln(k) \cdot \mathrm{cost}_1(\mathcal{U})$ and $M$ cuts is very likely to halve $c_{\max}(t)$ as in Lemma 2. It is thus very likely that the cost of these $M$ cuts upper bounds $\mathrm{cost\text{-}increase}(r)$. The additional constant factor of 2 in the statement of the lemma arises by considering the small "failure" probability of such a trial. The formal proof bounding the expected cost of this geometric series can be found in Appendix B.1 in the supplementary material.

### 3.1.1   Upper bounding cost by a factor of $O(\log^2 k)$

Observe that our analysis of Algorithm 1 implies that the expected cost of the output tree is at most $O(\log^2(k) \cdot \mathrm{cost}_1(\mathcal{U}))$ whenever $c_{\max}$ and $c_{\min}$ do not differ by more than a polynomial factor in $k$. However, our current techniques fail to upper bound this expectation by a factor $o(k)$ for general $c_{\max}$ and $c_{\min}$. To illustrate this point, consider the $k$-dimensional instance with a single point $\boldsymbol{x}$ at the origin and $k$ centers where the $i$-th center $\boldsymbol{\mu}^i$ is located at the $i$-th standard basis vector scaled by the factor $2^i$. In our analysis, we upper bound the cost of $\boldsymbol{x}$ with its *maximum* distance to those centers that remain in the same leaf whenever $\boldsymbol{x}$ is separated from its closest center $\boldsymbol{\mu}^1$. This yields the following upper bound on the expected cost of $\boldsymbol{x}$ in the final tree

$$\sum_{i=2}^{k} 2^i \Pr[\boldsymbol{x} \text{ is separated from } \boldsymbol{\mu}^1 \text{ and } \boldsymbol{\mu}^i \text{ is the farthest remaining center}].$$

Due to the exponentially increasing distances, this is lower bounded by $\sum_{i=2}^{k} 2^{i-1} \Pr[\boldsymbol{x} \text{ is separated from } \boldsymbol{\mu}^1 \text{ and } \boldsymbol{\mu}^i \text{ is a remaining center}]$. Now note that the probability in this last sum equals $\Pr[\boldsymbol{x} \text{ is separated from } \boldsymbol{\mu}^1 \text{ before } \boldsymbol{x} \text{ is separated from } \boldsymbol{\mu}^i] = 2/(2+2^i)$. It follows that any analysis of Algorithm 1 that simply upper bounds the reassignment cost of a point with the maximum distance to a remaining center cannot do better than a factor of $\Omega(k)$.

We overcome this obstacle by analyzing a slight modification of Algorithm 1 that avoids these problematic cuts that separate very close centers. Recall the notation used in the previous section: $\mathrm{Leaves}(t)$ denotes the state of Leaves at the beginning of the $t$-th iteration of the while loop and $c_{\max}(t) = \max_{B\in\mathrm{Leaves}(t)} \max_{\boldsymbol{\mu}^i,\boldsymbol{\mu}^j\in B} \|\boldsymbol{\mu}^i - \boldsymbol{\mu}^j\|_1$ denotes the maximum distance between two centers that belong to the same leaf at the beginning of the $t$-th iteration. We now modify Algorithm 1 by replacing Line 5 "Sample $(i, \theta)$ uniformly at random from AllCuts" by

Sample $(i, \theta)$ uniformly at random from those cuts in $\mathrm{AllCuts}$ that do *not* separate two centers that are within distance at most $c_{\max}(t)/k^4$.

This modification allows us to prove a nearly tight guarantee on the expected cost.

**Theorem 2.** *Given reference centers* $\mathcal{U} = \{\boldsymbol{\mu}^1, \boldsymbol{\mu}^2, \ldots, \boldsymbol{\mu}^k\}$, *modified Algorithm 1 outputs a threshold tree* $T$ *whose expected cost satisfies* $\mathbb{E}[\mathrm{cost}_1(T)] \leq O(\log^2 k) \cdot \mathrm{cost}_1(\mathcal{U})$.

The above theorem implies the furthermore statement of Theorem 1. This follows by observing that Algorithm 1 selects the same cuts as the modified version with probability at least $1 - 1/k$. This error probability can be made smaller by not allowing cuts that separate centers within distance $c_{\max}(t)/k^\ell$ for an $\ell$ larger than 4. Due to space limitations, we give a more formal explanation of this implication in Appendix B.2 and we prove Theorem 2 in Appendix B.3 in the supplementary material.

### 3.2 Implementation details

Since we only use cuts that split at least one leaf, the algorithm in fact only needs to sample cuts conditioned on this event. Note that if we sample only the cuts that split at least one leaf, the while loop in Line 4 of Algorithm 1 runs for at most $k - 1$ iterations. We now explain how to efficiently sample cuts (Line 5), find the leaves split by a given cut (Line 6), and implement the split operation (Lines 7–8).

We first show how to efficiently implement the split operation. For a cut $(i, \theta)$ that splits a given leaf $B$ into $B^-$ and $B^+$, the split operation can be implemented in $O(d \cdot \min(|B^-|, |B^+|) \cdot \log |B|)$ time as follows: In each leaf $B$, we maintain $d$ binary search trees $T_1^B, \ldots, T_d^B$ where $T_i^B$ stores the $i$-th coordinate of the centers in $B$. Now, given a cut $(i, \theta)$ and a leaf $B$ that gets split by $(i, \theta)$, we can find the number of centers in $B$ that have a smaller or equal $i$-th coordinate than $\theta$ using $T_i^B$ in $O(\log |B|)$ time. Let $b^-$ be this number, and let $b^+ = |B| - b^-$. Suppose that $b^- \leq b^+$. For the other case, the implementation is analogous. In this case, we construct $B^-$ by initializing it with $d$ empty binary search trees and inserting the centers whose $i$-th coordinate is at most $\theta$ to each of them. This takes $O(d \cdot b^- \cdot \log |B|)$ time. For $B^+$, we just reuse the binary search trees of $B$ after removing the centers that belong to $B^-$. This also takes $O(d \cdot b^- \cdot \log |B|)$ time. Let $\tau(k)$ denote the running time of all splitting operations performed by the algorithm when starting with a single leaf with $k$ leaves. Then, $\tau(k) = \tau(k - k') + \tau(k') + O(d \cdot \min(k - k', k') \cdot \log k)$ and by induction, we conclude that $\tau(k) = O(dk \log^2 k)$.

To find the leaves that get separated by a cut, we employ the following data structure. For each dimension $i$, we maintain a balanced interval tree $T_i^{\mathrm{int}}$. For each tentative leaf node with centers $B$, we store the interval indicating the range of the $i$-th coordinate of $B$ in $T_i^{\mathrm{int}}$. We update the corresponding interval trees after each split operation, which amounts to removing at most one interval and adding at most two intervals per node that gets split. Note that the added and removed intervals for a single split operation for a fixed dimension can be computed in $O(\log k)$ time using the previously described node binary search trees. Moreover, adding and removing intervals to and from an interval tree with at most $k$ intervals also takes $O(\log k)$ time. As we have $O(k)$ split operations in total, the time to maintain the interval trees is $O(dk \log k)$. Now, given a cut $(i, \theta)$, we can retrieve all the leaves that get separated by $(i, \theta)$ in $O(\log(k_1) + k_2)$ time where $k_1$ is the number of tentative leaves and $k_2$ is the number of tentative leaves that get separated by the cut. To retrieve such leaves, we query the $i$-th interval tree to find all intervals that contain the value $\theta$. Since we sample at most $k - 1$ cuts from the conditioned distribution, the total time for this operation over all cuts and all dimensions is $O(dk \log k)$.

What remains is to show that we can efficiently sample a uniform cut conditioned on the event that it splits at least one leaf. To this end, in the interval trees described above, we also maintain the lengths of the unions of intervals in each subtree. This length information can be updated in $O(\log k')$ time where $k' \leq k$ is the number of intervals in an interval tree. Then in $O(d)$ time, one can sample a dimension $i$ and in $O(\log k')$ time, sample a suitable $\theta$ value.

## 4 Explainable $k$-means and general $\ell_p$-norm clustering

In this section, we generalize Theorem 2 to the explainable $k$-clustering problems with assignment cost defined in terms of the $\ell_p$-norm, which includes the explainable $k$-means ($p = 2$) problem.

Recall that in Section 3, we sample cuts from the uniform distribution over AllCuts, and consequently, the probability that a point $\boldsymbol{x} \in \mathcal{X}$ is separated from its closest center $\pi(\boldsymbol{x})$ is proportional to the $\ell_1$ distance between $\boldsymbol{x}$ and $\pi(\boldsymbol{x})$. However, selecting cuts according to the uniform distribution can be arbitrarily bad for higher $p$-norms even in one-dimensional space. For example, consider the $k$-means (i.e. $p = 2$) problem with $d = 1$ where the cost of assigning a point $\boldsymbol{x}$ to a center $\boldsymbol{\mu}$ is defined as $\|\boldsymbol{x} - \boldsymbol{\mu}\|_2^2$. Suppose we have two centers $\boldsymbol{\mu}^1 = -1$ and $\boldsymbol{\mu}^2 = D > 1$, and fix one data point $\boldsymbol{x} = \boldsymbol{0}$. The closest center to $\boldsymbol{x}$ is $\boldsymbol{\mu}^1$ and hence the original cost is 1. However, the expected cost of a uniformly random cut is $((D \cdot 1^2 + 1 \cdot D^2)/(1 + D) = D$ which can be arbitrarily large.

To avoid such drastic costs, we sample cuts from a generalized distribution. Ideally, we would like to sample cuts analogously to the case of $k$-medians so that the probability that we separate a point $\boldsymbol{x}$ from its closest center $\pi(\boldsymbol{x})$ is proportional to $\|\boldsymbol{x} - \pi(\boldsymbol{x})\|_p^p$. However, sampling from such a distribution seems very complicated if at all possible. Instead, we sample from a slightly different distribution: Namely, for a $p$-norm where the cost of assigning a point $x$ to a center $y$ is $\|x - y\|_p^p$, we sample cuts $(i, \theta)$ from the distribution where the probability density function of $(i, \theta)$ is proportional to $\min_{j \in [k]} |\mu_i^j - \theta|^{p-1}$, the $(p-1)$-th power of the minimum distance to a center *along the $i$-th dimension*. We call this distribution $\mathcal{D}_p$.

Using samples from $\mathcal{D}_p$ with a modified version of Algorithm 1 yields Theorem 3. We defer the proof of Theorem 3 to Appendix C in the supplementary material.

**Theorem 3.** *For every $p \geq 1$, there exists a randomized algorithm that when given input centers $\mathcal{U} = \{\boldsymbol{\mu}^1, \boldsymbol{\mu}^2, \ldots, \boldsymbol{\mu}^k\}$, outputs a threshold tree $T$ whose expected cost satisfies*

$$\mathbb{E}[\mathrm{cost}_p(T)] \leq O(k^{p-1} \log^2 k) \cdot \mathrm{cost}_p(\mathcal{U}).$$

# 5   Lower bound

In this section we show how to construct an instance of the clustering problem such that any explainable clustering has cost at least $\Omega(k^{p-1})$ times larger than the optimal non-explainable clustering for the objective function given by $\ell_p$ norm, for every $p \geq 1$. In particular, for $p = 2$, this entails an $\Omega(k)$ lower bound for (explainable) $k$-means.

Let $m$ be a prime. Our hard instance is in $\mathbb{R}^d$ for $d = m \cdot (m - 1)$ and the set of dimensions corresponds to the set of all linear functions over $\mathbb{Z}_m$ with non-zero slope. That is, we associate the $i$-th dimension with the function $f_i : x \mapsto (a_i x + b_i) \bmod m$, where $a_i = 1 + \lfloor i/m \rfloor$ and $b_i = i \bmod m$. Consider $k = m$ centers $\boldsymbol{\mu}^1, \ldots, \boldsymbol{\mu}^k$ such that the $i$-th coordinate of the $j$-th center is given by $\mu_i^j = f_i(j)$. For each center $\boldsymbol{\mu}^j$ we create a set of $2d$ points $B_j$, each point differing from the center in exactly one dimension by either $-1$ or $+1$, i.e., $B_j = \{\boldsymbol{\mu}^j + c \cdot \boldsymbol{e}^i \mid c \in \{-1, 1\}, i \in [d]\}$, where $\boldsymbol{e}^i$ denotes the standard basis vector in the $i$-th dimension. Then, our hard instance is just $\bigcup_{j \in [k]} B_j$.

Since every point is at distance exactly 1 from its closest center, the cost of the optimal clustering OPT is equal to the total number of points $n = 2dk$ (regardless of the $\ell_p$ norm). We prove, in Appendix D in the supplementary material, that:

Claim 1.  Any two centers are at the same distance $\delta = \Theta(d^{1/p}k)$ from each other.

Claim 2.  Any nontrivial threshold cut, i.e., one that separates some two centers, separates also some two points from the same $B_j$.

It follows that, in any explainable clustering, already the first threshold cut (from the decision tree's root) forces some two points from the same set $B_j$ to eventually end up in two different leaves, and hence at least one of the $k$ leaves has to contain two points from two different $B_j$'s. The distance between these two points, by the triangle inequality, is at least $\delta - 2$, and therefore the cost of the explainable clustering is at least $\Omega(\delta^p) = \Omega(dk^p)$, which is $\Omega(k^{p-1}) \cdot$ OPT.

## Acknowledgments and Disclosure of Funding

This research was supported by the Swiss National Science Foundation projects 200021-184656 "Randomness in Problem Instances and Randomized Algorithms" and 185030 "Lattice Algorithms and Integer Programming."

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
