$\mu$ is defined as $\|x - \mu\|_2^2$. Suppose we have two centers $\mu^1 = -1$ and $\mu^2 = D > 1$, and fix one data

point $x = 0$. The closest center to $x$ is $\mu^1$ and hence the original cost is 1. However, the expected cost of a uniformly random cut is $((D \cdot 1^2 + 1 \cdot D^2)/(1 + D) = D$ which can be arbitrarily large.

To avoid such drastic costs, we sample cuts from a generalized distribution. Ideally, we would like to sample cuts analogously to the case of $k$-medians so that the probability that we separate a point $x$ from its closest center $\pi(x)$ is proportional to $\|x - \pi(x)\|_p^p$. However, sampling from such a distribution seems very complicated if at all possible. Instead, we sample from a slightly different distribution: Namely, for a $p$-norm where the cost of assigning a point $x$ to a center $y$ is $\|x - y\|_p^p$, we sample cuts $(i, \theta)$ from the distribution where the probability density function of $(i, \theta)$ is proportional to $\min_{j \in [k]} |\mu_i^j - \theta|^{p-1}$, the $(p - 1)$-th power of the minimum distance to a center *along the $i$-th dimension*. We call this distribution $\mathcal{D}_p$.

Using samples from $\mathcal{D}_p$ with a modified version of Algorithm 1 yields Theorem 3. We defer the proof of Theorem 3 to Appendix C.

**Theorem 3.** *For every $p \geq 1$, there exists a randomized algorithm that when given input centers $\mathcal{U} = \{\mu^1, \mu^2, \ldots, \mu^k\}$, outputs a threshold tree $T$ whose expected cost satisfies*

$$\mathbb{E}[\text{cost}_p(T)] \leq O(k^{p-1} \log^2 k) \cdot \text{cost}_p(\mathcal{U}).$$

## 5   Lower bound

In this section we show how to construct an instance of the clustering problem such that any explainable clustering has cost at least $\Omega(k^{p-1})$ times larger than the optimal non-explainable clustering for the objective function given by $\ell_p$ norm, for every $p \geq 1$. In particular, for $p = 2$, this entails an $\Omega(k)$ lower bound for (explainable) $k$-means.

Let $m$ be a prime. Our hard instance is in $\mathbb{R}^d$ for $d = m \cdot (m - 1)$ and the set of dimensions corresponds to the set of all linear functions over $\mathbb{Z}_m$ with non-zero slope. That is, we associate the $i$-th dimension with the function $f_i : x \mapsto (a_i x + b_i) \mod m$, where $a_i = 1 + \lfloor i/m \rfloor$ and $b_i = i \mod m$. Consider $k = m$ centers $\mu^1, \ldots, \mu^k$ such that the $i$-th coordinate of the $j$-th center is given by $\mu_i^j = f_i(j)$. For each center $\mu^j$ we create a set of $2d$ points $B_j$, each point differing from the center in exactly one dimension by either $-1$ or $+1$, i.e., $B_j = \{\mu^j + c \cdot e^i \mid c \in \{-1, 1\}, i \in [d]\}$, where $e^i$ denotes the standard basis vector in the $i$-th dimension. Then, our hard instance is just $\bigcup_{j \in [k]} B_j$.

Since every point is at distance exactly 1 from its closest center, the cost of the optimal clustering OPT is equal to the total number of points $n = 2dk$ (regardless of the $\ell_p$ norm). We prove, in Appendix D, that:

Claim 1.  Any two centers are at the same distance $\delta = \Theta(d^{1/p} k)$ from each other.

Claim 2.  Any nontrivial threshold cut, i.e., one that separates some two centers, separates also some two points from the same $B_j$.

It follows that, in any explainable clustering, already the first threshold cut (from the decision tree's root) forces some two points from the same set $B_j$ to eventually end up in two different leaves, and hence at least one of the $k$ leaves has to contain two points from two different $B_j$'s. The distance between these two points, by the triangle inequality, is at least $\delta - 2$, and therefore the cost of the explainable clustering is at least $\Omega(\delta^p) = \Omega(dk^p)$, which is $\Omega(k^{p-1}) \cdot$ OPT.

### Acknowledgments and Disclosure of Funding

This research was supported by the Swiss National Science Foundation projects 200021-184656 "Randomness in Problem Instances and Randomized Algorithms" and 185030 "Lattice Algorithms and Integer Programming."

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

# A    The minimum cut algorithm loses $\Omega(k)$ factor for $k$-medians

We give an example where the minimum cut algorithm of [6] produces a threshold tree with cost $\Omega(k)$ times the cost of an optimal clustering in the $\ell_1$-norm. The idea is to start with the lower bound example in Section 5 since any two centers are "far apart". By adding a dimension for each center in which fewer edges are cut, the minimum cut will make linearly many cuts that split only one center. Combined with the large distance to reassign a point to the wrong center, the result is the minimum cut algorithm losing an $\Omega(k)$ factor. In the $\ell_1$ norm, it suffices to map half of the coordinate values to -1 and the other half to +1 and still maintain the "large" distance between centers. The remainder of this section is a formal description of the instance.

Take the lower bound example from Section 5 and increase the dimension by $k$. Now the points are in $\mathbb{R}^{d+k}$ with $d + k$ coordinates (recall that $d = m(m - 1)$ and $k = m$ with $m$ prime). First, we describe the $k$ centers $\mathcal{U}' = \{\boldsymbol{\mu}'^1, \ldots, \boldsymbol{\mu}'^k\}$ as a mapping from the centers $\mathcal{U} = \{\boldsymbol{\mu}^1, \ldots, \boldsymbol{\mu}^k\}$ in Section 5. For the first $d$ coordinates, $\mu_i'^j = \mu_i^j \mod 2$. For the last $k$ coordinates, center $\boldsymbol{\mu}'^j$ has a 0 in every coordinate $d + i$, $1 \leq i \leq k$, except coordinate $d + j$ which is a 1.

The reasoning behind this mapping is that the family of functions $f_i$ in Section 5 is the standard construction of a family of pairwise independent hash functions [8]. In particular, if $f_{a,b}(x) = (ax + b) \mod m$ and $h_{a,b}(x) = f_{a,b}(x) \mod 2$, then for $x \neq y$, $h_{a,b}(x) = h_{a,b}(y)$ with probability at most $1/2$ when $a$ and $b$ are chosen uniformly at random from $\{0, 1, \ldots, m-1\}$, $a \neq 0$. Recall that $f_i(x) = (a_i x + b_i) \mod m$ where $a_i, b_i$ range over all elements in $\{1, 2, \ldots, m-1\}, \{0, \ldots, m-1\}$, respectively, and $\mu_i^j = f_i(j)$. Fix any pair of centers $\boldsymbol{\mu}^{j_1}, \boldsymbol{\mu}^{j_2}$ where $j_1 \neq j_2$. Note that picking $i \in [d]$ uniformly at random is equivalent to picking $a_i, b_i \in \{0, 1, \ldots, m - 1\}$, $a \neq 0$, uniformly at random due to the definition of $a_i, b_i$. We have $\mu_i'^{j_1} = (\mu_i^{j_1} \mod 2) = (\mu_i^{j_2} \mod 2) = \mu_i'^{j_2}$ with probability at most $1/2$ over the uniformly random choice of $i$, so any pair of centers are the same on at most $1/2$ of the coordinates. Hence, our new centers $\mathcal{U}'$ are at pairwise distance $\Theta(d)$.

Now we define the remaining points. Let $\boldsymbol{e}^i$ be the standard $(d + k)$-dimensional $i$-th basis vector. Similar to Section 5, we have a set $B'_j$ for each center $\boldsymbol{\mu}'^j$ with $2d$ points where each point differs from $\boldsymbol{\mu}'^j$ on one of the first $d$ coordinates by $\pm 1$. Additionally, we want $(k - 1)/2$ points to differ on one of the last $k$ coordinates. To this end, define $B'_j = \{\boldsymbol{\mu}'^j + c \cdot \boldsymbol{e}^i \mid c \in \{-1, 1\}, i \in [d]\} \cup \{\boldsymbol{\mu}'^j - \boldsymbol{e}^{d+j}\}^{(k-1)/2}$, where $\{\boldsymbol{\mu}'^j - \boldsymbol{e}^{d+j}\}^{(k-1)/2}$ denotes a multiset of $(k - 1)/2$ copies of the point $\boldsymbol{\mu}'^j - \boldsymbol{e}^{d+j}$.

In particular, our construction has the following properties:

1. The distance between any pair of centers is $\Theta(d)$.

2. A cut $(i, \theta)$ in any dimension $i$, $1 \leq i \leq d$, and $\theta \in (0, 1)$ splits some two centers and the number of points separated is equal to the number of centers.

3. A cut $(i, \theta)$ in any dimension $i$, $d + 1 \leq i \leq d + k$, and $\theta \in (0, 1)$ splits some two centers and separates $\approx k/2$ points.

Property (2) holds because for any dimension $1 \leq i \leq d$ and for each center $\boldsymbol{c}$, $c_i$ is either 1, in which case there is exactly one point at $\boldsymbol{c} - \boldsymbol{e}^i$, or 0, in which case there is exactly one point at $\boldsymbol{c} + \boldsymbol{e}^i$. Note that this further implies that, when (after separating some centers) $x$ centers are remaining, the number of points separated by a cut of type (2) will be equal to $x$. Then the cuts in (3) will be minimum for $\approx k/2$ cuts of all minimum cuts required to separate all centers since each cut in (3) separates exactly one center from the remaining centers. Hence we have that the minimum cut algorithm of [6] will construct a threshold tree with $\Omega(k)$ height by making some $\Omega(k)$ cuts in dimensions $d + 1$ through $d + k$.

To see that the minimum cut algorithm loses a $\Omega(k)$ factor, note that the optimal clustering has a value of $2dk + (k - 1)k/2 = \Theta(k^3)$. The first term in the sum is because each of $2d$ points in each cluster differs from the center by $\pm 1$ in exactly one of the first $d$ coordinates and the second term is because $(k - 1)/2$ of the points in each cluster differ by $-1$ from the center in one of the last $k$ coordinates. On the other hand, an algorithm that always makes a minimum cut incurs a cost of $\Theta(dk^2)$ to reassign $\approx k/2$ points to the wrong center for $\approx k/2$ centers, just for those cuts of type (2). This gives an overall cost of $\Omega(dk^2)$ for the threshold tree produced. Since $d = \Theta(k^2)$ we have that the minimum cut algorithm is $\Omega(k)$ away from the cost of an optimal clustering.

# B Omitted proofs of Section 3

## B.1 Upper bounding cost increase of a round

Here we give the formal proof of Lemma 3, restated below. Recall that

$$\text{cost-increase}(r) = \sum_{t \,:\, c_{\max}(t) \in (c_{\max}/2^{r+1}, c_{\max}/2^r]} c_{\max}(t) f_{i_t}(\theta_t)$$

is the random variable that upper bounds the cost increase caused by the cuts selected during the iterations $t$ when $c_{\max}/2^{r+1} < c_{\max}(t) \leq c_{\max}/2^r$.

**Lemma 3.** *For every $r$, $\mathbb{E}[\text{cost-increase}(r)] \leq 12 \ln(k) \cdot \text{cost}_1(\mathcal{U})$.*

*Proof.* Let $t$ be the first iteration when $c_{\max}(t) \leq c_{\max}/2^r$ and let $M = 3 \ln(k) \cdot 2L/c_{\max}(t)$ as in Lemma 2. In the following, we use $\text{cost}_M$ to denote the random variable that equals the cost increase caused by adding $M$ uniformly random cuts after the $t$-th iteration. Then

$$\mathbb{E}[\text{cost}_M] \leq M \cdot c_{\max}(t) \cdot \mathbb{E}_{(i,\theta)}[f_i(\theta)] \leq M \cdot c_{\max}(t) \cdot \text{cost}_1(\mathcal{U})/L = 6 \ln(k) \cdot \text{cost}_1(\mathcal{U}) \,,$$

where the first inequality holds because $c_{\max}(t)$ is monotonically decreasing and the second inequality is by Lemma 1. At the same time, if we let $H$ denote the event that $c_{\max}(t)$ has halved after adding these $M$ cuts, i.e., that $c_{\max}(t + M) \leq c_{\max}(t)/2$, then $\Pr[H] \geq 1 - 1/k$ by Lemma 2. We now upper bound the expectation of $\text{cost-increase}(r)$ by considering "trials" of $M$ cuts until one of these succeeds in halving $c_{\max}(t)$. Indeed, split the sequence of random cuts selected by the algorithm after iteration $t$ into such trials $A_1, \ldots, A_\ell$ where each $A_j$ consist of $M$ cuts, and $A_\ell$ is the first successful trial in the sense that selecting (only) those cuts after iteration $t$ would cause $c_{\max}(t)$ to halve. Then we must have that $c_{\max}(t)$ has halved also after adding all the cuts in the $\ell$ trials. It follows that $\text{cost-increase}(r)$ is upper bounded by the cost increase caused by the cuts in $A_1, A_2, \ldots, A_\ell$. We can thus upper bound $\mathbb{E}[\text{cost-increase}(r)]$ by the expected cost of these trials until one succeeds:

$$\sum_{i=0}^{\infty} \Pr[H] \cdot \Pr[\neg H]^i \cdot \left( \mathbb{E}\left[\text{cost}_M \mid H\right] + i \cdot \mathbb{E}\left[\text{cost}_M \mid \neg H\right] \right) \,,$$

where we use $\mathbb{E}[\text{cost}_M \mid H]$ and $\mathbb{E}[\text{cost}_M \mid \neg H]$ for the expected costs of a successful and unsuccessful trials, respectively. By standard calculations (as for the geometric distribution), this upper bound simplifies to $\mathbb{E}\left[\text{cost}_M \mid H\right] + \frac{\Pr[\neg H]}{\Pr[H]} \mathbb{E}\left[\text{cost}_M \mid \neg H\right]$. This can be further rewritten as

$$\frac{1}{\Pr[H]} \cdot \left( \Pr[H] \cdot \mathbb{E}[\text{cost}_M \mid H] + \Pr[\neg H] \cdot \mathbb{E}[\text{cost}_M \mid \neg H] \right) = \frac{\mathbb{E}[\text{cost}_M]}{\Pr[H]} \leq 12 \ln(k) \cdot \text{cost}_1(\mathcal{U}) \,,$$

where we used that $\Pr[H] \geq 1 - 1/k \geq 1/2$. $\qquad\square$

## B.2 Theorem 2 implies furthermore statement of Theorem 1

Recall that the difference between Algorithm 1 and the modified version is that Algorithm 1 samples cuts uniformly at random whereas the modified version only adds a random cut if it does *not* separate two centers that are within distance $c_{\max}(t)/k^4$.

Algorithm 1 adds $k - 1$ cuts to its tree. We now argue that these $k - 1$ cuts are with probability at least $1 - 1/k$ sampled from the same distribution as the $k - 1$ cuts added by the modified version. This then implies the furthermore statement of Theorem 1 since Theorem 2 says that the expected cost of the modified algorithm is $O(\log^2 k) \cdot \text{cost}_1(\mathcal{U})$. To this end, consider the $i$-th such cut and let $t$ be the iteration when the $(i-1)$-st cut was added to the tree. Then when the $i$-th cut is added there must be two centers in the same leaf at distance $c_{\max}(t)$. So the probability that two centers within distance $c_{\max}(t)/k^4$ are separated by the $i$-th cut (which is a uniformly random cut among all cuts that would separate at least two centers in the same leaf) is at most $1/k^4$. There can be at most $\binom{k}{2}$ such pairs and so by the union bound, we can conclude that, with probability at least $1 - 1/k^2$, the $i$-th cut of Algorithm 1 does not separate any such nearby centers. We can thus view the distribution from which Algorithm 1 samples the $i$-th cut as follows: With probability $p \leq 1/k^2$ it samples a uniformly random cut that separates two centers within distance $c_{\max}(t)/k^4$ and with remaining

probability it samples a uniformly random cut that does not separate any such centers, i.e., from the same distribution that the modified algorithm samples the $i$-th cut from. Applying the union bound over the $k-1$ cuts then yields the furthermore statement of Theorem 1. Finally, we remark that the same arguments imply a larger success probability if applied to the modified algorithm that only adds cut that do not separate centers within distance $c_{\max}(t)/k^\ell$ for some $\ell \geq 4$.

## B.3  Upper bounding expected cost of modified Algorithm 1

We prove Theorem 2, i.e., that modified Algorithm 1 returns a threshold tree whose expected cost is $O(\log^2 k) \cdot \mathrm{cost}_1(\mathcal{U})$. The proof is similar to the cost analysis in Section 3.1 with the main difference being that here we are more careful in bounding the cost when considering different "rounds" of the algorithm. In the analysis it will be convenient to take the following viewpoint of the modified algorithm: it samples a uniformly random cut and then discards it if it separates two centers within distance $c_{\max}(t)/k^4$. While the number of iterations may increase with this viewpoint, the output distribution is the same as the modified algorithm in that, in each iteration, a cut is sampled uniformly at random among those that do not separate any centers within distance $c_{\max}(t)/k^4$. In the following, we refer to this as the *sample-discard algorithm* and we prove Theorem 2 by showing that the sample-discard algorithm outputs a tree whose expected cost is $O(\log^2 k) \cdot \mathrm{cost}_1(\mathcal{U})$.

Let $(i_t, \theta_t)$ denote the (uniformly random) cut selected in the $t$-th iteration of the sample-discard algorithm and recall the following notation: $\mathrm{Leaves}(t)$ denotes the state of Leaves at the beginning of the $t$-th iteration of the while-loop and $c_{\max}(t) = \max_{B \in \mathrm{Leaves}(t)} \max_{\boldsymbol{\mu}^i, \boldsymbol{\mu}^j \in B} \|\boldsymbol{\mu}^i - \boldsymbol{\mu}^j\|_1$ denotes the maximum distance between two centers that belong to the same leaf at the beginning of the $t$-th iteration. We start by observing that Lemma 2 readily generalizes to the modified version.

**Lemma 4.** *Fix the the threshold cuts selected by the sample-discard algorithm during the first $t-1$ iterations (this determines the random variable $\mathrm{Leaves}(t)$ and thus $c_{\max}(t)$). Let $M = 3\ln(k) \cdot 4L/c_{\max}(t)$. Then*

$$\Pr[c_{\max}(t+M) \leq c_{\max}(t)/2] \geq 1 - 1/k \,,$$

*where the probability is over the random cuts selected in iterations $t, t+1, \ldots, t+M-1$.*

*Proof.* The proof is similar to that of Lemma 2 but some care has to be taken as certain cuts are now discarded.

Consider two centers $\boldsymbol{\mu}^i$ and $\boldsymbol{\mu}^j$ that belong to the same leaf in $\mathrm{Leaves}(t)$. Further suppose that $c_{\max}(t)/2 \leq \|\boldsymbol{\mu}^p - \boldsymbol{\mu}^q\|_1 \leq c_{\max}(t)$. We have that any cut $(i, \theta)$ that separates these two centers is considered (i.e., not discarded) by the sample-discard algorithm after the $t$-th iteration unless $(i, \theta)$ also separates two centers within distance $c_{\max}(t)/k^4$. Here we used that $c_{\max}(\cdot)$ is monotonically decreasing and so the set of cuts that are discarded if sampled is only decreasing in later iterations. We can thus obtain the lower bound $c_{\max}(t)/(4L)$ on the probability that a uniformly random cut separates $\boldsymbol{\mu}^p$ and $\boldsymbol{\mu}^q$ by subtracting

$$\frac{1}{L} \sum_{i=1}^{d} \int_{-\infty}^{\infty} \mathbb{1}\left[ \theta \text{ separates two centers within distance } c_{\max}(t)/k^4 \right] d\theta \leq \frac{1}{L} \binom{k}{2} \frac{c_{\max}(t)}{k^4}$$

from

$$\frac{1}{L} \sum_{i=1}^{d} \int_{-\infty}^{\infty} \mathbb{1}[\theta \text{ between } \boldsymbol{\mu}_i^p \text{ and } \boldsymbol{\mu}_i^q] \, d\theta \geq \frac{1}{L} c_{\max}(t)/2 \,.$$

The proof now proceeds in the exact same way as that of Lemma 2. Indeed, if the centers are at distance at least $c_{\max}(t)/2$, the probability that they are *not* separated by any of $M$ independently chosen cuts is at most

$$\left(1 - \frac{c_{\max}(t)}{4L}\right)^M = \left(1 - \frac{c_{\max}(t)}{4L}\right)^{3\ln(k) \cdot 4L/c_{\max}(t)} \leq (1/e)^{3\ln(k)} = 1/k^3 \,.$$

There are at most $\binom{k}{2}$ pairs of centers in the leaves of $\mathrm{Leaves}(t)$ at distance at least $c_{\max}(t)/2$. By the union bound, we thus have, with probability at least $1 - 1/k$, that each of these pairs are separated by at least one of the cuts selected in iterations $t, t+1, \ldots, t+M-1$. In that case, any two centers in the same leaf of $\mathrm{Leaves}(t+M)$ are at distance at most $c_{\max}(t)/2$ and so $c_{\max}(t+M) \leq c_{\max}(t)/2$. $\square$

Now, similar to Section 3.1, we can upper bound the expected cost of the constructed threshold tree $T$ by

$$\mathbb{E}\left[\text{cost}_1(T)\right] \leq \text{cost}_1(\mathcal{U}) + \mathbb{E}\left[\sum_t c_{\max}(t) f_{i_t}(\theta_t) \mathbb{1}[(i_t, \theta_t) \text{ was added to the tree}]\right],$$

where the sum is over the iterations. We remark that, in contrast to Section 3.1, we have strengthened the upper bound by only considering those cuts that were actually added to the threshold tree by the modified algorithm. This refinement is necessary for obtaining the improved guarantee. We now analyze the sum in the expectation by partitioning it into $1 + \lfloor \log_2(c_{\max}/c_{\min}) \rfloor$ rounds. Specifically for $r \in \{0, 1 \ldots, \lfloor \log_2(c_{\max}/c_{\min}) \rfloor\}$, we let

$$\text{cost-increase}'(r) = \sum_{t:c_{\max}(t)\in(c_{\max}/2^{r+1}, c_{\max}/2^r]} c_{\max}(t) f_{i_t}(\theta_t) \mathbb{1}[(i_t, \theta_t) \text{ was added to the tree}]$$

be the cost of the cuts selected during the iterations $t$ when $c_{\max}/2^{r+1} < c_{\max}(t) \leq c_{\max}/2^r$.

To upper bound $\mathbb{E}[\text{cost-increase}'(r)]$ we use $\text{active}_r(i, \theta) \in \{0, 1\}$ to denote the indicator variable of those cuts that separate two centers within distance $c_{\max}/2^r$ and do not separate any two centers within distance $c_{\max}/(2^{r+1}k^4)$.

**Lemma 5.** *For a round $r$,*

$$\mathbb{E}[\text{cost-increase}'(r)] \leq 24 \ln(k) \cdot \sum_{i=1}^d \int_{-\infty}^\infty f_i(\theta) \, \text{active}_r(i, \theta) \, d\theta.$$

Before giving the proof of this lemma, let us see how it implies Theorem 2. For this, note that a cut $(i, \theta)$ only has $\text{active}_r(i, \theta) = 1$ for at most $O(\log(k^4))$ many values of $r$. Indeed, let $c$ be the distance between the closest centers that $(i, \theta)$ separates. Then any round $r$ for which $\text{active}_r(i, \theta) = 1$ must satisfy $c_{max}/(2^{r+1}k^4) \leq c \leq c_{max}/2^r$. Hence, we have

$$\text{cost}_1(T) \leq \text{cost}_1(\mathcal{U}) + \sum_r \mathbb{E}[\text{cost-increase}'(r)]$$

$$\leq \text{cost}_1(\mathcal{U}) + \sum_r 24 \ln(k) \cdot \sum_{i=1}^d \int_{-\infty}^\infty f_i(\theta) \, \text{active}_r(i, \theta) \, d\theta$$

$$\leq \text{cost}_1(\mathcal{U}) + O(\log^2 k) \cdot \sum_{i=1}^d \int_{-\infty}^\infty f_i(\theta) \, d\theta$$

$$= O(\log^2 k) \cdot \text{cost}_1(\mathcal{U}).$$

In other words, we proved that the sample-discard algorithm outputs a tree $T$ with $\mathbb{E}[\text{cost}_1(T)] \leq O(\log^2 k) \cdot \text{cost}_1(\mathcal{U})$, which implies Theorem 2 since modifed Algorithm 1 and the sample-discard algorithm have the same output distribution. It remains to prove the lemma.

*Proof of Lemma 5.* Consider the first iteration $t$ such that $c_{\max}(t) \leq c_{\max}/2^r$. Further suppose that $c_{\max}(t) > c_{\max}/2^{r+1}$ since otherwise $\text{cost-increase}'(r) = 0$ and the statement is trivial. We proceed to upper bound $\mathbb{E}[\text{cost-increase}'(r)]$ as follows. First note that the cost of a random cut sampled in an iteration $t'$ such that $c_{\max}/2^{r+1} \leq c_{\max}(t') \leq c_{\max}(t)$ equals

$$\frac{c_{\max}(t')}{L} \sum_{i=1}^d \int_{-\infty}^\infty f_i(\theta) \mathbb{1}[(i, \theta) \text{ was added to the tree}] \, d\theta.$$

The cut $(i, \theta)$ can be added to the tree only if it does not separate any centers within distance $c_{\max}(t')/k^4 \geq c_{\max}/(2^{r+1}k^4)$ and it must separate two centers within distance at most $c_{\max}(t') \leq c_{\max}/2^r$. In other words, any cut that is added to the tree must have $\text{active}_r(i, \theta) = 1$. We can thus upper bound the above cost of a single cut by

$$\frac{c_{\max}(t)}{L} \sum_{i=1}^d \int_{-\infty}^\infty f_i(\theta) \, \text{active}_r(i, \theta) \, d\theta, \tag{2}$$

where we also used that $c_{\max}(t') \le c_{\max}(t)$.

The proof now follows arguments that are again similar to those in Section 3.1. Select $M = 12 \ln(k) \cdot L/c_{\max}(t)$ as in Lemma 4. We upper bound $\mathbb{E}[\text{cost-increase}(r)]$ by adding "trials" of $M$ cuts until $c_{\max}(\cdot)$ goes below $c_{\max}/2^{r+1}$. (Strictly speaking this may not happen after a multiple of $M$ cuts but considering more cuts may only increase the cost of our upper bound.) Let $H$ be the event that the following $M$ cuts causes $c_{\max}(\cdot)$ to drop below $c_{\max}/2^{r+1}$. By Lemma 4, $\Pr[H] \ge 1 - 1/k$. Furthemore, the expected cost of $M$ cuts is $M$ times (2) which equals

$$12 \ln(k) \cdot \sum_{i=1}^{d} \int_{-\infty}^{\infty} f_i(\theta) \, \text{active}_r(i, \theta) \, d\theta.$$

The statement now follows from the same "geometric distribution" calculations as in the proof of Lemma 3. $\qquad\square$

## C  Omitted proofs of Section 4

We now prove Theorem 3 (restated below for convenience) by showing how to modify Algorithm 1 to get the desired guarantee on the expected cost.

**Theorem 3.** *For every $p \ge 1$, there exists a randomized algorithm that when given input centers $\mathcal{U} = \{\boldsymbol{\mu}^1, \boldsymbol{\mu}^2, \dots, \boldsymbol{\mu}^k\}$, outputs a threshold tree $T$ whose expected cost satisfies*

$$\mathbb{E}[\text{cost}_p(T)] \le O(k^{p-1} \log^2 k) \cdot \text{cost}_p(\mathcal{U}).$$

To prove Theorem 3, we consider a generalized version of Algorithm 1 where we sample threshold cuts from the distribution $\mathcal{D}_p$ introduced in Section 4. Recall that $\mathcal{D}_p$ is defined such that the p.d.f. of a cut $(i, \theta)$ is proportional to the $(p-1)$-th power of the minimum distance from $\theta$ to a projection of a center in the $i$-th dimension.

We start by introducing some notation and making the definition of $\mathcal{D}_p$ precise. For a dimension $i \in [d]$, let $\mu_i^- = \min_{j \in [k]} \mu_i^j$ and $\mu_i^+ = \max_{j \in [k]} \mu_i^j$. For a dimension $i \in [d]$ and two coordinates $x, y \in \mathbb{R}$, let $\mathcal{I}_i(x, y)$ be the set of consecutive intervals along the $i$-th dimension delimited by the coordinates $x$ and $y$ themselves and the projections of the centers in $\mathcal{U}$ that lie between $x$ and $y$. For example, consider the 2-dimensional instance with four centers $\boldsymbol{\mu}^1, \dots, \boldsymbol{\mu}^4$ shown in Fig. 3. On the horizontal axis, two coordinates $x$ and $y$ are marked along with the projections of the four centers $\mu_1^1, \mu_1^2, \mu_1^3,$ and $\mu_1^4$. Here, $\mathcal{I}_1(x, y)$ consists of the three consecutive intervals $[x, \mu_1^4], [\mu_1^4, \mu_1^2],$ and $[\mu_1^2, y]$.

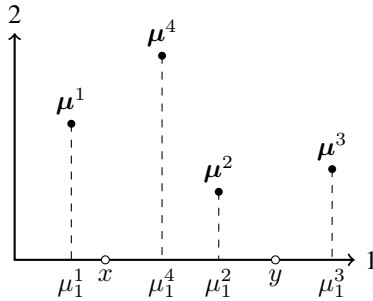

Figure 3: Intervals defined by projecting points onto a coordinate axis.

Observe that, by the definition of $\mathcal{I}_i(x, y)$, we have $|x - y| = \sum_{[a,b] \in \mathcal{I}_i(x,y)} |b - a|$.

Let

$$\mathcal{I}_{\text{all}} = \bigcup_{i \in [d]} \left\{ (i, [a, b]) : [a, b] \in \mathcal{I}_i(\mu_i^-, \mu_i^+) \right\}$$

denote the collection of all dimension–interval pairs that are delimited by the projections of the centers onto the respective dimensions. We define

$$L_p = \sum_{(i, [a,b]) \in \mathcal{I}_{\text{all}}} |b - a|^p.$$

With the introduced notation, the distribution $\mathcal{D}_p$ can be formally described as follows: We first select a dimension $i$ and an interval $[a, b] \in \mathcal{I}_i(\mu_i^-, \mu_i^+)$ along with dimension $i$ (i.e., we select a dimension–interval pair $(i, [a, b]) \in \mathcal{I}_{\text{all}}$) with probability $|b - a|^p / L_p$. Then we pick $\theta \in [a, b]$ randomly such that the p.d.f. $\theta$ is

$$P_{a,b}(\theta) := \frac{p \cdot 2^{p-1}}{(b-a)^p} \min(\theta - a, b - \theta)^{p-1}.$$

Another key component of the design and analysis of the generalized algorithm is a *pseudo-distance* function. For two points $\boldsymbol{x}, \boldsymbol{y} \in \mathbb{R}^d$, Let

$$\mathcal{I}(\boldsymbol{x}, \boldsymbol{y}) = \bigcup_{i \in [d]} \{(i, [a, b]) : [a, b] \in \mathcal{I}_i(x_i, y_i)\}.$$

We then define the pseudo-distance between $\boldsymbol{x}$ and $\boldsymbol{y}$ as

$$d_p(\boldsymbol{x}, \boldsymbol{y}) = \sum_{(i, [a,b]) \in \mathcal{I}(\boldsymbol{x}, \boldsymbol{y})} |b - a|^p.$$

Note that the $p$-th power of the $\ell_p$ distance, $\|\boldsymbol{x} - \boldsymbol{y}\|_p^p$, between two points $\boldsymbol{x}$ and $\boldsymbol{y}$ is defined as $\sum_{i \in [d]} |x_i - y_i|^p$. It is easy to see that $\|\boldsymbol{x} - \boldsymbol{y}\|_p^p \geq d_p(\boldsymbol{x}, \boldsymbol{y})$ since

$$|x_i - y_i|^p = \left( \sum_{[a,b] \in \mathcal{I}_i(x_i, y_i)} |a - b| \right)^p \geq \sum_{[a,b] \in \mathcal{I}_i(x_i, y_i)} |a - b|^p$$

for each dimension $i$. For $p = 1$, we have equality.

A key observation now is that, if we sample a cut from $\mathcal{D}_p$, the probability that it separates two centers $\boldsymbol{\mu}^g$ and $\boldsymbol{\mu}^h$ is *proportional to their pseudo-distance $d_p(\boldsymbol{\mu}^g, \boldsymbol{\mu}^h)$*.

## C.1 The algorithm for $\ell_p$-norms with $p \geq 1$.

We now present the generalized algorithm. The only difference from the modified version of Algorithm 1 is how we sample random cuts at Line 5. Recall from Section 3 that we defined Leaves($t$) to denote the state of Leaves at the beginning of the $t$-th iteration. We define $c'_{p,\max}(t)$ as the maximum pseudo-distance between any pair of centers in a leaf in Leaves($t$). Formally, $c'_{p,\max}(t) = \max_{B \in \text{Leaves}(t)} \max_{\boldsymbol{\mu}^i, \boldsymbol{\mu}^j \in B} d_p(\boldsymbol{\mu}^i, \boldsymbol{\mu}^j)$. Let $c'_{p,\max} = c'_{p,\max}(1)$.

Now, in the sampling step (Line 5), we draw samples from $\mathcal{D}_p$. However, we discard the cut if it separates any two centers in a leaf whose pseudo-distance is at most $c'_{p,\max}(t)/k^4$. Note that this is a generalization of the sample-discard algorithm from Appendix B.3. We present the pseudo-code in Algorithm 2.

---

**Algorithm 2:** Generalized explainable clustering algorithm for higher $\ell_p$-norms.

---

1 **Input:** A collection of $k$ centers $\mathcal{U} = \{\boldsymbol{\mu}^1, \boldsymbol{\mu}^2, \ldots, \boldsymbol{\mu}^k\} \subset \mathbb{R}^d$.
2 **Output:** A threshold tree with $k$-leaves.
3 Leaves $\leftarrow \{\mathcal{U}\}$
4 **while** $|\text{Leaves}| < k$ **do**
5 $\quad$ Sample a cut $(i, \theta)$ from $\mathcal{D}_p$
6 $\quad$ **if** $(i, \theta)$ *separates two centers that are closer than* $c'_{p,\max}(\cdot)/k^4$ *in pseudo-distance* **then**
7 $\quad\quad$ Discard the cut.
8 $\quad$ **else**
9 $\quad\quad$ **for** *each* $B \in$ Leaves *that are split by* $(i, \theta)$ **do**
10 $\quad\quad\quad$ Split $B$ into $B^-$ and $B^+$ and add them as left and right children of $B$.
11 $\quad\quad\quad$ Update Leaves.

12 **return** the threshold tree defined by all cuts that separated some $B$.

---

Following the lines of Appendix B.3, we now upper bound the expected cost of Algorithm 2.

**Lemma 6.** *Fix the threshold cuts selected by Algorithm 2 during the first $t-1$ iterations. Let $M = 3 \cdot 4 \cdot \ln(k) \cdot L_p / c'_{p,max}(t)$. Then*

$$\Pr[c'_{p,\max}(t+M) \le c'_{p,\max}(t)/2] \ge 1 - 1/k,$$

*where the probability is over the random cuts selected in iterations $t, t+1, \ldots, t+M-1$.*

*Proof.* We begin by introducing a few more notations that are useful in the analysis. For an iteration $t$, let $\mathrm{TooClose}(t)$ be the set of pairs of centers $(\boldsymbol{\mu}^g, \boldsymbol{\mu}^h)$ that satisfy $d_p(\boldsymbol{\mu}^g, \boldsymbol{\mu}^h) \le c'_{p,\max}(t)/k^4$. In other words, $\mathrm{TooClose}(t)$ contains pairs of centers that the algorithm is not allowed to separate at the $t$-th iteration. Note that for any $(\boldsymbol{\mu}^g, \boldsymbol{\mu}^h) \in \mathrm{TooClose}(t)$, both $\boldsymbol{\mu}^g$ and $\boldsymbol{\mu}^h$ will be in the same leaf in $\mathrm{Leaves}(t)$. Let

$$\mathcal{I}_{\mathrm{bad}}(t) = \bigcup_{(\boldsymbol{\mu}^g, \boldsymbol{\mu}^h) \in \mathrm{TooClose}(t)} \mathcal{I}(\boldsymbol{\mu}^g, \boldsymbol{\mu}^h)$$

be the set of dimension–interval pairs $(i, [a, b])$ such that making a cut in interval $[a, b]$ along dimension $i$ will separate a pair of centers in TooClose. Observe that a cut that is made outside of $\mathcal{I}_{\mathrm{bad}}(t)$ will not separate any pair of centers in TooClose.

Consider a leaf $B \in \mathrm{Leaves}(t)$ and two centers $\boldsymbol{\mu}^g$ and $\boldsymbol{\mu}^h$ in $B$ such that $c'_{p,\max}(t)/2 \le d_p(\boldsymbol{\mu}^g, \boldsymbol{\mu}^h) \le c'_{p,\max}(t)$.

Note that

$$\sum_{[a,b] \in \mathcal{I}_{\mathrm{bad}}(t)} |b-a|^p \le \sum_{(\boldsymbol{\mu}^{g'}, \boldsymbol{\mu}^{h'}) \in \mathrm{TooClose}(t)} \sum_{(i,[a,b]) \in \mathcal{I}(\boldsymbol{\mu}^{g'}, \boldsymbol{\mu}^{h'})} |b-a|^p$$

$$= \sum_{(i,[a,b]) \in \mathcal{I}(\boldsymbol{\mu}^{g'}, \boldsymbol{\mu}^{h'})} d_p(\boldsymbol{\mu}^{g'}, \boldsymbol{\mu}^{h'})$$

$$\le \binom{k}{2} \frac{c'_{p,\max}(t)}{k^4} \le \frac{c'_{p,\max}(t)}{4}.$$

In the last inequality, we use that $k \ge 2$.

Hence, the probability that a cut selected at the $t$-th iteration separates $\boldsymbol{\mu}^g$ and $\boldsymbol{\mu}^h$ is at least

$$\frac{d_p(\boldsymbol{\mu}^g, \boldsymbol{\mu}^h)}{L_p} - \frac{\sum_{[a,b] \in \mathcal{I}_{\mathrm{bad}}(t)} |b-a|^p}{L_p} \ge \frac{c'_{p,\max}(t)}{2L_p} - \frac{c'_{p,\max}(t)}{4L_p} \ge \frac{c'_{p,\max}(t)}{4L_p}.$$

The proof now follows by replacing $c_{\max}(t)$ with $c'_{p,\max}(t)$ and $L$ with $L_p$ in the remaining part of the proof of Lemma 4. $\qquad \square$

In the following analysis, we use the Hölder's inequality stated below:

**Lemma 7** (Hölder's inequality)**.** *For two real numbers $u$ and $v$ such that $1/u + 1/v = 1$ and two positive real number sequences $y_1, \ldots, y_m$ and $z_1, \ldots, z_m$, it holds that*

$$\sum_{i \in [m]} y_i z_i \le \left( \sum_{i \in [m]} y_i^u \right)^{1/u} \left( \sum_{i \in [m]} z_i^v \right)^{1/v}.$$

*In particular, setting $y_1 = y_2 = \cdots = y_m = 1$, $u = p/(p-1)$ and $v = p$ for some $p$, and taking the $p$-th power on both sides, it holds that*

$$\left( \sum_{i \in [m]} z_i \right)^p \le m^{p-1} \sum_{i \in [m]} z_i^p.$$

We now upper bound the expected cost. Recall that $\pi(\boldsymbol{x})$ denotes the closest center in $\mathcal{U}$ to a point $\boldsymbol{x} \in \mathcal{X}$ and that $\mathrm{cost}_p(\mathcal{U})$ is defined as

$$\mathrm{cost}_p(\mathcal{U}) = \sum_{\boldsymbol{x} \in \mathcal{X}} \|\boldsymbol{x} - \pi(\boldsymbol{x})\|_p^p = \sum_{\boldsymbol{x} \in \mathcal{X}} \sum_{i \in [d]} |x_i - \pi(\boldsymbol{x})_i|^p.$$

To bound the cost of the output clustering in the $k$-medians setting, we used the triangle inequality. For general $p$-th power of $p$-norms, we use the following generalized triangle inequality:

**Lemma 8.** *Consider three points $x, y, z \in \mathbb{R}^d$. We have $\|z - x\|_p^p \le 2^{p-1} \left( \|z - y\|_p^p + \|y - x\|_p^p \right)$.*

*Proof.* Expanding $\| \cdot \|_p^p$ as a summation over $d$ dimensions, it is sufficient to prove that for any three real numbers $x, y, z \in \mathbb{R}$, $|z - x|^p \le 2^{p-1}(|z - y|^p + |y - x|^p)$. Without loss of generality, assume that $z \ge x$. If $y \le x$ or $y \ge z$, the proof follows trivially because we have $|z - x| \le |z - y|$ or $|z - x| \le |y - z|$, respectively. Now suppose that $x \le y \le z$. Let $a = y - x$ and $b = z - y$. Since $a + b = z - x$, we simply need to prove that $(a + b)^p \le 2^{p-1}(a^p + b^p)$ which follows from Hölder's inequality. $\square$

Recall that we defined $c'_{p,\max}(t)$ and $c'_{p,\max}$ earlier using the pseudo-distance function $d_p$. We now define $c_{p,\max}(t)$ and $c_{p,\max}$ similarly, but using the $p$-th power of the $\ell_p$ norm: Namely, $c_{p,\max}(t) = \max_{B \in \text{Leaves}(t)} \max_{\mu^i, \mu^j \in B} \|\mu^i - \mu^j\|_p^p$ and $c_{p,\max} = c_{p,\max}(1)$. We again use $(i_t, \theta_t)$ to denote the cut selected by Algorithm 2 in the $t$-th iteration and $f_i(\theta)$ to denote the number of points $x \in \mathcal{X}$ that are separated from $\pi(x)$ by a cut $(i, \theta)$.

For a point $x \in \mathcal{X}$, suppose that it is assigned to some center $\mu$ in the final threshold tree. If $\mu = \pi(x)$, the cost contribution of $x$ in the final clustering is the same as that in the original clustering. Suppose $\mu \ne \pi(x)$ and suppose that $x$ was separated from $\pi(x)$ at iteration $t$. Then, using Lemma 8, we conclude that the cost of assigning $x$ to $\mu$, i.e., $\|x - \mu\|_p^p$, is upper bounded by

$$2^{p-1} \left( \|x - \pi(x)\|_p^p + \| \pi(x) - \mu\|_p^p \right) \le 2^{p-1} \left( \|x - \pi(x)\|_p^p + c_{p,\max}(t) \right).$$

Let UsedCuts be the set of cuts used to split some leaf in Line 10 of Algorithm 2. Now using the above observation, we can upper bound the expected cost of the output tree, $\mathbb{E}[\text{cost}_p(T)]$, by

$$\text{cost}_p(T) \le 2^{p-1} \left( \text{cost}_p(\mathcal{U}) + \sum_{r=0}^{\infty} \mathbb{E}[\text{cost-increase}(r)] \right)$$

where

$$\text{cost-increase}(r) = \sum_{t: \frac{c'_{p,\max}}{2^{r+1}} \le c'_{p,\max}(t) \le \frac{c'_{p,\max}}{2^r}} c_{p,\max}(t) \cdot f_{i_t}(\theta_t) \cdot \mathbb{1}[(i_t, \theta_t) \in \text{UsedCuts}].$$

Note that in the last expression, the summed terms use $c_{p,\max}(t)$ whereas the condition of the summation uses $c'_{p,\max}(t)$. Note that

$$\text{cost-increase}(r) \le \sum_{t: \frac{c'_{p,\max}}{2^{r+1}} \le c'_{p,\max}(t) \le \frac{c'_{p,\max}}{2^r}} k^{p-1} c'_{p,\max}(t) \cdot f_{i_t}(\theta_t) \cdot \mathbb{1}[(i_t, \theta_t) \in \text{UsedCuts}]$$

$$\le k^{p-1} \frac{c'_{p,\max}}{2^r} \cdot \sum_{t: \frac{c'_{p,\max}}{2^{r+1}} \le c'_{p,\max}(t) \le \frac{c'_{p,\max}}{2^r}} f_{i_t}(\theta_t) \cdot \mathbb{1}[(i_t, \theta_t) \in \text{UsedCuts}].$$

The first inequality is by Hölder's inequality (applied independently in each dimension in the computation of respective $d_p$ and $\| \cdot \|_p^p$ values). The second inequality simply uses the condition of the summation.

We now upper bound the expected value of cost-increase$(r)$. Let $\mathcal{I}_{\text{act}}(r)$ be the set of dimension–interval pairs in $\mathcal{I}$ that do not separate any pair of centers that are closer than $c'_{p,\max}/(k^4 2^{r+1})$ in pseudo-distance but separate at least one pair of centers that are closer than $c'_{p,\max}/2^r$ in pseudo-distance. We prove the following lemma which is analogous to Lemma 5 in Appendix B.3.

**Lemma 9.** *For a round $r$, $\mathbb{E}[\text{cost-increase}(r)]$ is*

$$O \left( k^{p-1} \cdot \log k \right) \cdot \left( \sum_{(i, [a,b]) \in \mathcal{I}_{\text{act}}(r)} |b - a|^p \int_a^b P_{a,b}(\theta) f_i(\theta) d\theta \right).$$

*Proof.* We consider "trials" of $M$ consecutive iterations in round $r$ where

$$M = 12 \cdot 2^{r+1} \ln(k) \cdot L_p / c'_{p,max}.$$

We perform independent trials until $c'_{p,\max}(\cdot)$ at the end of a trial goes below $c'_{p,\max}/2^{r+1}$.

Consider one trial and let $s$ be the starting iteration of the trial. Note that we have

$$M \geq 3 \cdot 4 \cdot \ln(k) \cdot L_p / c'_{p,\max}(s)$$

since $c'_{p,\max}(s) \geq c'_{p,\max}/2^{r+1}$. Thus, by Lemma 6, after $M$ iterations, round $r$ ends with probability at least $1 - 1/k \geq 1/2$. (Note that round $r$ may end before all M iterations of a trial are completed. In such trials, we assume that we discard the additional cuts that are made after the round ends.)

Let $\mathrm{UB}_\ell = \sum_{s=t'}^{t'+M-1} f_{i_s}(\theta_s) \cdot \mathbb{1}[(i_s, \theta_s) \in \mathrm{UsedCuts}]$ and observe that

$$\text{cost-increase}(r) \leq k^{p-1} \frac{c'_{p,\max}}{2^r} \cdot \sum_\ell \mathrm{UB}_\ell \tag{3}$$

where the sum is over all trials we perform in round $r$.

We first upper bound each term $\mathrm{UB}_\ell$ and then use the expectation of a geometric random variable to upper bound the expected value of $\sum_\ell \mathrm{UB}_\ell$. We have

$$\mathbb{E}[\mathrm{UB}_\ell] \leq \sum_{s=t'}^{t'+M-1} \mathbb{E}\left[f_{i_s}(\theta_s) \cdot \mathbb{1}[(i_s, \theta_s) \in \mathrm{UsedCuts}]\right]$$

$$\leq \sum_{s=t'}^{t'+M-1} \frac{1}{L_p} \sum_{(i,[a,b]) \in \mathcal{I}_{\mathrm{act}}(r)} |b-a|^p \int_a^b P_{a,b}(\theta) f_i(\theta) d\theta. \tag{4}$$

Note that we only sum over dimension–interval pairs in $\mathcal{I}_{\mathrm{act}}(r)$ as cuts made outside of this set will be discarded. To elaborate, the dimension–interval pair in which a cut $(i, \theta)$ is made can be outside of $\mathcal{I}_{\mathrm{act}}(r)$ for two reasons:

1. Because it separates two centers that are closer than $c'_{p,\max}/(k^4 2^{r+1}) \leq c'_{p,\max}(t')/k^4$. Then it will get discarded in Line 7.

2. Because it does not separate two centers that are closer than $c'_{p,\max}/2^r$. Such a cut will not split any leaves in Line 9.

Consequently, for all $(i_s, \theta_s) \in \mathrm{UsedCuts}$, we have $\theta_s \in [a, b]$ for some interval $[a, b]$ such that $(i_s, [a, b]) \in \mathcal{I}_{\mathrm{act}}(r)$.

Now, since the summed terms in (4) no longer depend on the summed index $s$, we now have

$$\mathbb{E}[\mathrm{UB}_\ell] \leq \frac{M}{L_p} \sum_{(i,[a,b]) \in \mathcal{I}_{\mathrm{act}}} |b-a|^p \int_a^b P_{a,b}(\theta) f_i(\theta) d\theta$$

$$= \frac{12 \cdot 2^{r+1} \ln(k)}{c'_{p,\max}} \sum_{(i,[a,b]) \in \mathcal{I}_{\mathrm{act}}} |b-a|^p \int_a^b P_{a,b}(\theta) f_i(\theta) d\theta.$$

Now, considering that round $r$ ends after a trial with probability at least $1/2$, using the expected value of a geometric distribution, we conclude that

$$\mathbb{E}\left[\sum_\ell \mathrm{UB}_\ell\right] \leq \frac{24 \cdot 2^{r+1} \ln(k)}{c'_{p,\max}} \sum_{(i,[a,b]) \in \mathcal{I}_{\mathrm{act}}(r)} |b-a|^p \int_a^b P_{a,b}(\theta) f_i(\theta) d\theta.$$

The proof of the lemma then follows by combining this with the bound in (3). $\square$

With Lemma 9 in hand, we now prove Theorem 3.

*Proof of Theorem 3.* Using Lemma 9, we can upper bound $\sum_{r=0}^{\infty} \mathbb{E}[\text{cost-increase}(r)]$ as follows:

$$\sum_{r=0}^{\infty} \mathbb{E}[\text{cost-increase}(r)]$$

$$= O\left(k^{p-1} \cdot \log k\right) \cdot \sum_{r=0}^{\infty} \sum_{(i,[a,b]) \in \mathcal{I}_{\text{act}}(r)} |b-a|^p \int_a^b P_{a,b}(\theta) f_i(\theta) d\theta$$

$$= O\left(k^{p-1} \cdot \log k\right) \cdot \sum_{(i,[a,b]) \in \mathcal{I}} \sum_{r=0}^{\infty} \mathbb{1}[(i,[a,b]) \in \mathcal{I}_{\text{act}}(r)] \cdot \left(|b-a|^p \int_a^b P_{a,b}(\theta) f_i(\theta) d\theta\right)$$

$$= O\left(k^{p-1} \cdot \log k\right) \cdot \sum_{(i,[a,b]) \in \mathcal{I}} |b-a|^p \int_a^b P_{a,b}(\theta) f_i(\theta) d\theta \cdot \left(\sum_{r=0}^{\infty} \mathbb{1}[(i,[a,b]) \in \mathcal{I}_{\text{act}}(r)]\right).$$

We now claim that for any dimension–interval pair in $(i,[a,b]) \in \mathcal{I}$

$$\sum_{r=0}^{\infty} \mathbb{1}[(i,[a,b]) \in \mathcal{I}_{\text{act}}(r)] = \left|\{r : (i,[a,b]) \in \mathcal{I}_{\text{act}}(r)\}\right| = O(\log k). \tag{5}$$

Namely, fix some dimension–interval pair $(i,[a,b]) \in \mathcal{I}$. Let $c$ be the smallest pseudo-distance between any pair of centers that are separated if a cut $(i,\theta)$ such that $\theta \in [a,b]$ is made. Then $(i,[a,b])$ is in $\mathcal{I}_{\text{act}}(r)$ only if $c \leq c'_{p,\max}/2^r$ and $c'_{p,\max}/2^{r+1} \leq k^4 c$, or equivalently, $\log(c'_{p,\max}/2ck^4) \leq r \leq \log(c'_{p,\max}/c)$ which yields (5). Thus we have

$$\sum_{r=0}^{\infty} \mathbb{E}[\text{cost-increase}(r)]$$

$$= O\left(k^{p-1} \cdot \log^2 k\right) \cdot \sum_{(i,[a,b]) \in \mathcal{I}} \left(|b-a|^p \int_a^b P_{a,b}(\theta) f_i(\theta) d\theta\right)$$

$$= O\left(k^{p-1} \cdot \log^2 k\right) \cdot \sum_{(i,[a,b]) \in \mathcal{I}} \left(|b-a|^p \int_a^b p2^{p-1} \frac{\min(\theta-a, b-\theta)^{p-1}}{|b-a|^p} f_i(\theta) d\theta\right)$$

$$= O\left(k^{p-1} \cdot \log^2 k\right) \cdot (p2^{p-1}) \cdot \sum_{(i,[a,b]) \in \mathcal{I}} \int_a^b \min(\theta-a, b-\theta)^{p-1} f_i(\theta) d\theta. \tag{6}$$

Now to conclude the proof of Theorem 3, let $\text{cost}'_p(\mathcal{U}) = \sum_{\boldsymbol{x} \in \mathcal{X}} d_p(\boldsymbol{x}, \pi(\boldsymbol{x}))$ which is the cost of $\mathcal{U}$ defined in terms of the pseudo-distances. Recall that $\|\boldsymbol{x} - \boldsymbol{y}\|_p^p \geq d_p(\boldsymbol{x}, \boldsymbol{y})$ and hence we have $\text{cost}_p(\mathcal{U}) \geq \text{cost}'_p(\mathcal{U})$ where the equality holds if $p = 1$. We then have

$$\text{cost}_p(\mathcal{U}) \geq \text{cost}'_p(\mathcal{U}) = \sum_{\boldsymbol{x} \in \mathcal{X}} d_p(\boldsymbol{x}, \pi(\boldsymbol{x})) = \sum_{\boldsymbol{x} \in \mathcal{X}} \sum_{i \in [d]} \sum_{[a,b] \in \mathcal{I}_i(x_i, \pi(x)_i)} |a-b|^p$$

$$= \sum_{\boldsymbol{x} \in \mathcal{X}} \sum_{i \in [d]} \sum_{[a,b] \in \mathcal{I}_i(x_i, \pi(x)_i)} \int_a^b p(\theta-a)^{p-1} d\theta$$

$$\geq \sum_{\boldsymbol{x} \in \mathcal{X}} \sum_{(i,[a,b]) \in \mathcal{I}} \int_a^b p \cdot \min(\theta-a, b-\theta)^{p-1} \cdot \mathbb{1}[\theta \text{ is between } x_i \text{ and } \pi(x)_i] \, d\theta \tag{7}$$

$$= p \sum_{(i,[a,b]) \in \mathcal{I}} \int_a^b \min(\theta-a, b-\theta)^{p-1} f_i(\theta) \, d\theta. \tag{8}$$

The inequality in (7) above needs an explanation. First notice that, by the definition of $\mathcal{I}$, summing over $(i,[a,b]) \in \mathcal{I}$ is the same as summing over $i \in [d]$ and $[a,b] \in \mathcal{I}_i(\mu_i^-, \mu_i^+)$. Fix some point $\boldsymbol{x}$ and dimension $i$, and w.l.o.g. assume that $x_i \leq \pi(\boldsymbol{x})_i$. Then each interval $[a,b] \in \mathcal{I}_i(\mu_i^-, \mu_i^+)$ falls into one of the following categories:

1. $b \leq x_i$ or $\pi(\boldsymbol{x}))_i \leq a$. The contribution from such intervals are zero in both sides of the inequality in (7).

2. $x_i \leq a \leq b \leq \pi(\boldsymbol{x})_i$. The contribution from such intervals to the left hand side of (7) is at least as the contribution to the right hand side because of the $\min$ function.

3. $a < x_i < b$. Note that $\int_a^b (\theta - a)^{p-1} d\theta = \int_a^b (b - \theta)^{p-1} d\theta$. Hence, in this case, the contributions to both sides of (7) are equal if $x_i \geq (a+b)/2$. Otherwise, the contribution to the L.H.S. is higher.

4. $a < \pi(\boldsymbol{x}) < b$. This case is analogous to Item 3.

The inequality in (7) follows by applying this observation to each interval in $\mathcal{I}$ and each point in $\mathcal{X}$.

The theorem statement then follows by combining bounds (6) and (8). $\qquad\square$

### C.2 Implementation details

Note that Algorithm 2 differs from Algorithm 1 in the sampling step and the new sample discarding step. Recall that the implementation details of Algorithm 1 is presented in Section 3.2. In this section, assuming that we can sample a $\theta \in [a, b]$ with p.d.f. $P_{a,b}(\theta)$ from a given interval $[a, b]$ in $\Theta(1)$ time, we show how to efficiently implement the sampling step of Algorithm 2. In particular, we show how to select the dimension–interval pair $(i, [a, b])$ with probability proportional to $|b - a|^p$. Once the cut is sampled, the discarding step can be implemented by simulating the splitting operation and ignoring the cut if it separates two centers that are too close.

Suppose that for each dimension $i$, we maintain a data structure $S_i$ that stores the intervals in $\mathcal{I}_i(\mu_i^-, \mu_i^+)$ that are not yet split by a cut. There are $k - 1$ disjoint intervals in $\mathcal{I}_i(\mu_i^-, \mu_i^+)$, and we assume they are ordered by the left coordinate and indexed $[a_1, b_1], \ldots, [a_{k-1}, b_{k-1}]$. Additionally, each $S_i$ supports the following operations:

1. Initialize with all intervals in $\mathcal{I}_i(\mu_i^-, \mu_i^+)$ in $O(k \log k)$ time.

2. Remove an interval in $\mathcal{I}_i(\mu_i^-, \mu_i^+)$ in $O(\log k)$ time.

3. Given two indices $\ell, r \in [k-1]$, answer the query for $\sum_{j=\ell}^{r} |b_j - a_j|^p \mathbb{1}[(a_j, b_j) \in S_i]$ in $O(\log k)$ time.

We can implement $S_i$ as a segment tree.

Now we can sample an interval from $S_1, \ldots, S_d$ as follows in $O(d \log k)$ time: We first query $(1, k-1)$ in each tree, aggregate the results, and pick a dimension $i$ with the correct probability. Then we select an interval from $S_i$ with the correct probability by employing a binary-search like algorithm. To elaborate, we first query it for $(1, \lfloor (k-1)/2 \rfloor)$ and $(\lfloor (k-1)/2 \rfloor) + 1, k-1)$ and use the results to randomly decide if the index of the sampled interval should be in the sub-range $\{1, \ldots, \lfloor (k-1)/2 \rfloor\}$ or $\{\lfloor (k-1)/2 \rfloor) + 1, \ldots, k-1\}$. Then we recursively apply the same procedure on the selected sub-range of indices until we end up with only one interval. A crude runtime analysis gives $O(\log^2 k)$ running time for the recursive sampling as there are $O(\log k)$ queries and each query takes $O(\log k)$ time. However we can modify the segment tree such that the partial sums maintained in the segment tree coincide with our queries so that each query can be answered in constant time.

## D  Omitted proofs of Section 5

*Proof of Claim 1.* Fix two different centers $\boldsymbol{\mu}^{j_1}$, $\boldsymbol{\mu}^{j_2}$, $j_1 \neq j_2$. Their distance $\delta$ satisfies

$$\delta^p = \sum_{i \in [d]} \left| f_i(j_1) - f_i(j_2) \right|^p = \sum_{a=1}^{m-1} \sum_{b=0}^{m-1} \left| (aj_1 + b) \bmod m - (aj_2 + b) \bmod m \right|^p.$$

For $a \in \{1, \ldots, p-1\}$, let $x(a) = (aj_1 - aj_2) \bmod m$. Observe that

$$\left| (aj_1 + b) \bmod m - (aj_2 + b) \bmod m \right| \in \{x(a), m - x(a)\},$$

and whether it is $x(a)$ or $m - x(a)$ depends on $b$, with it being $x(a)$ for exactly $m - x(a)$ values of $b$ and $m - x(a)$ for the remaining $x(a)$ values of $b$. Hence,

$$\delta^p = \sum_{a=1}^{m-1} (m - x(a)) \cdot x(a)^p + x(a) \cdot (m - x(a))^p.$$

Since $j_1 \not\equiv j_2 \pmod{m}$, we have $\{x(a) \mid a \in \{1, \ldots, m-1\}\} = \{1, \ldots, m-1\}$, and

$$\delta^p = \sum_{i=1}^{m-1} (m - i) \cdot i^p + i \cdot (m - i)^p = 2 \cdot \sum_{i=1}^{m-1} (m - i) \cdot i^p = \Theta(m^{p+2}) = \Theta(dk^p).$$

$\square$

*Proof of Claim 2.* Let the cut be $(i, \theta)$. It must be that $0 \leq \theta < m - 1$, because otherwise the cut would not separate any two centers. Note that there exists a center $\boldsymbol{\mu}^j$ with $\mu_i^j = \lfloor \theta \rfloor$. Indeed, consider $j = (\lfloor \theta \rfloor - b_i) \cdot a_i^{-1} \bmod m$, using the fact that $a_i$ and $m$ are coprime. To finish the proof observe that the cut separates point $(\boldsymbol{\mu}^j + \boldsymbol{e}^i) \in B_j$ from all other points in $B_j$. $\square$