# OpenReview forum: "Nearly-Tight and Oblivious Algorithms for Explainable Clustering"
_NeurIPS.cc/2021/Conference — NeurIPS 2021 Poster_

### Official Review · Reviewer_65No · 2021-07-06

**Rating:** 7
**Confidence:** 4

**Summary:**

Given a set of points X in R^d, an explainable k-clustering of X is a partition of X in k sets that can be defined by a decision tree whose generic node has the form "x_i > t_i ?" where x_i is the i-th feature and t_i is a threshold value. The work studies the problem of computing explainable clusterings that are good in the sense of k-means and k-medians. The main results is a randomized algorithm that, given the centers of any k-clustering of X (not necessarily explainable), computes a novel k-clustering that is explainable and whose cost is only O(log^2 k) times worse that of the input in the case of k-medians, and O(k log^2 k) times worse in the case of k-means. The work gives near-matching lower bounds as well.


**Limitations And Societal Impact:**

Limitations are adequately discussed. There is no foreseeable societal impact.



**Main Review:**

TLDR: I think this is a good paper with clear (and clean) results, and deserves acceptance.

Originality: good. The results, including the algorithm, are original work. The authors compare correctly against previous work.

Quality: good. I read only some of the proofs, and they look ok to me. The authors point out both the strengths and the weaknesses of the work.

Clarity: good. The paper is clearly written and I found it very smooth to read. There are only a couple of points that perhaps could be improved, but these are minor concerns.

Significance: good. The paper advances the state of the art, in at least three ways: (1) it gives dimension-free bounds on the "cost of explainability" that are significantly better than the existing ones, (2) it shows that those bounds are almost optimal, (3) it shows that those bounds can be achieved by a very simple algorithm (I checked the previous work, and that algorithm is much more intricate).


**Time Spent Reviewing:**

4

---

### Official Review · Reviewer_gBnK · 2021-07-08

**Rating:** 6
**Confidence:** 3

**Summary:**

The paper suggests an algorithm that aims to understand how models make their decisions. In particular, the paper handles clustering approaches, including $k$-means and $k$-medians, and aims to explain their behavior by using the notion of "explainability" using a special instance of a decision tree, called threshold decision trees, where each edge in the tree is made with respect to a single dimension using a threshold on its value.

I enjoyed reading the paper! However, please see my comments below.

**Limitations And Societal Impact:**

The authors have mentioned two limitations of their work, which are interesting. The first which the authors have mentioned is that their upper bound on the cost increase of assigning a single point to a wrong center is not tight, whereas the second, concerns their analysis that it may include the cost of the same point multiple times. My main concern which was not addressed in the paper is that the theoretical guarantees involve expectation. My concern is, what is the worst-case analysis of your approximation that is obtained via "explainability" using either Algorithm 1 or its modified version?

**Main Review:**

In this paper, the author suggests two algorithms, one which is naive in nature with theoretical guarantees in obtaining an explainable clustering concerning the $k$-median and $k$-means problems.
While the first algorithm is fairly simple, however, the approximation can be very large, especially when the diameter of the set of centers is very large (exponential in the number of points, for example), its modified version handles this case by discarding cuts that will separate between a pair of centers having "small" distance between them.

 I believe the theoretical results are correct. I have looked into most of the proofs though not all, due to time constraints.

Minor Comments:
- I think that the left side of Equation (1) is not correct! Did you mean to write $\left\lVert \boldsymbol{x} - \pi(\boldsymbol{x}) + \pi(\boldsymbol{x}) - \mu\right\rVert_1$?
- Where there is $O$ notation in any of the inequalities/equalities you use, please use $\in$ instead for mathematical correctness.

Comment:

* I wonder whether it is better to apply importance sampling on the cuts than using uniform sampling. Coresets are known to approximate given loss functions concerning a set of queries (or all possible queries). In addition, they shine a light on the underlying structure of the data which in terms, aids in understanding the probability distribution from which the points were obtained. Since clustering aims in encapturing such hidden structures, the centers will encapture such hidden information within themselves (in the structure of their coordinates, for example, their distance from each other, etc.).  Using such information, one can sample better cuts with higher probability which will help in attaining tighter upper bounds than those achieved by uniform sampling. In light of this, why did you choose to go with uniform sampling? Can you give a brief explanation of whether importance sampling would be a better candidate than uniform sampling?


* See my concerns with the limitations of your work.


**Time Spent Reviewing:**

10

---

> ### Author Response · Authors · 2021-08-09
> **Response to Review of Paper4474 by Reviewer gBnK**
>
> Response:
> There is indeed a typo in Equation (1), thank you for pointing that out. In the submitted version one vertical bar is missing, we meant to write ||x - \pi(x)||_1 + ||\pi(x) - \mu||_1. As explained before the equation, this value upper bounds ||x - \mu||_1 by the triangle inequality (applied to x, \pi(x), and \mu).
>
> We are not sure if we understood correctly your concern regarding expectation in the guarantee, and in particular what you mean by the worst-case analysis. Let us stress that our guarantee holds for worst-case inputs, and the expectation is only over the internal random bits generated by the algorithm. When desired, there is a standard method to convert the expectation bound to a with-high-probability bound -- by Markov inequality the cost is below twice the expectation with probability at least 0.5; repeating the algorithm O(log n) times and taking the cheapest clustering yields the desired bound w.h.p. -- that however requires looking at the n data points to evaluate the clustering cost, while obtaining a clustering with good expected cost can be done with our algorithm by looking at only k centers. Actually, one can see that it is impossible to get very strong concentration bounds without looking at the points: take two centers located at (0,0, …, 0) and (1,1,1,..., 1) and a single point with coordinates (0, …, 0, 1, 0, …, 0) i.e., all 0’s except a single 1 in one out of the d dimensions. Now if we select this dimension at random, then any algorithm that doesn’t look at the points will output a solution of value d with probability at least 1/d whereas the cost of the initial clustering was 1.
>
> Regarding importance sampling, we actually conjecture that our algorithm is tight (at least to constant factors) for explainable k-median clustering. We therefore think uniform sampling is the answer for worst-case guarantees. However, with assumptions on the data distribution, other distributions may be better. This is an interesting topic for future research and it seems non-trivial to find the right assumptions as the lower bounds are in fact very “clusterable” instances, in the traditional usage of these concepts.

---

### Official Review · Reviewer_fBhF · 2021-07-14

**Rating:** 7
**Confidence:** 4

**Summary:**

The authors study the important problem of explainable clustering presenting new algorithms that improve substantially over the  state of the art results by shaving a k factor for k-medians and a klog^2 factor for k-means. The new results are polylog away from optimal. The algorithms are easy to implement and very efficient and the authors present also a lowerbound.



**Limitations And Societal Impact:**

See main review.

The main limitation is the lack of experimental evaluation.

Suggestions:
“Moreover, this algorithm is 115 oblivious to the data points, which avoids introducing biases.” -> I suggest to remove this as the bias can be in how the solution is used by the algorithm. The proof hinges on showing a relationship between the optimum cost and the number of points separated by random cuts as well as a bound of the increase of the cost for each separation.

Minor comments:

 “52 As a consequence, the algorithm cannot overfit the data (any more than the reference clustering 53 possibly already does), and the same expected cost guarantees hold for any future data points not 54 known at the time of the clustering construction” Please explain why future data points have this guarantee as it Is not obvious here.

**Main Review:**

Nearly-Tight and Oblivious Algorithms for Explainable Clustering

The authors study the important problem of explainable clustering as defined by Moshkovitz et al. The authors present a new algorithm which improves substantially over the previous state of the art results by shaving a k factor for k-medians and a klog^2 factor for k-means. The new results are polylog away from optimal.
The algorithms presented are also easy to implement and very efficient. Remarkably the algorithm only needs to look at the output of a non-exaplainable clustering (i.e. O(dk) input space).
The authors also show an improved lowerbound.

More precisely technically the algorithm is based on running a simple procedure that samples uniformly cuts from the bounding box of the centers in input until a valid tree is formed.

The only drawback of the paper is the complete lack of an empirical evaluation. This is quite surprising for a paper that shows  a simple and efficient algorithm, as it would not be difficult at all to test it. This leaves the possibility that the algorithm, in practice, actually underperforms the state of the art methods.  This would not be completely surprising as the algorithm is based on oblivious randomized cuts so it may actually be not great (on average instances) while being close to optimal on worst-case instances. I personally think that an empirical evaluation is warranted and the lack of it is lowers the value of the paper.

Technically the algorithm is based on interesting observations on the problem, and obtaining a tighter upper bound on the optimal cost of reassignment. Similar techniques are used for k-means. Overall the technical material is good.

All in all the paper is quite strong and I would give a strong accept but for the lack of empirical evaluation.


**Time Spent Reviewing:**

1.5

---

> ### Author Response · Authors · 2021-08-09
> **Response to Review of Paper4474 by Reviewer fBhF**
>
> Thank you for your time in reviewing our paper.
>
> We agree that an empirical evaluation of our algorithm would be interesting but we also think that a thorough empirical evaluation is beyond the scope of this paper. Indeed, our motivation in this work is to improve our theoretical understanding of the problem, just as the original paper by Moshkovitz et al. does, which does not contain an empirical evaluation.

---

> > ### Comment · Reviewer_fBhF · 2021-08-19
> > **Thanks**
> >
> > Thanks for the reply I agree that the empirical work is not need for this submission

---

### Official Review · Reviewer_mWXa · 2021-07-20

**Rating:** 8
**Confidence:** 4

**Summary:**

The paper improves the results of Dasgupta et al. for explainable clustering, under the $k$-median and $k$-means objectives (and also $\ell_p$-norms). In this problem, clustering solutions are restricted to partitions achievable via a collection of single coordinate cuts (where the coordinates are assumed to be features that explain the resulting clustering). The cost of a resulting clustering is measured against a reference (non-explainable) clustering that the algorithm takes into account. The algorithms in this paper are remarkably simple (and I mean that in a good way), and involve iterating over selecting coordinate cuts at random, subject to the next cut refining the separation between the reference cluster centers.

**Ethical Concerns:**

None.

**Limitations And Societal Impact:**

There are no immediate concrete applications, so societal impact is mostly irrelevant.

**Main Review:**

The improvement is substantial (from $k$ to $\log^2 k$ for $k$-median, and from $k^2$ to $k\log^2 k$ for $k$-means), and nearly matches lower bounds (a previous one in the case of $k$-median and a new one from this paper in the case of $k$-means). The algorithm is simple and attractive. The paper is clearly written and easy to follow. Overall, this is a nice theoretical contribution.

**Time Spent Reviewing:**

4 hours

---

### Decision · Program_Chairs · 2021-09-28

**Decision:**

Accept (Poster)

**Comment:**

The reviewers provided sufficient positive feedback and support for this paper, valuing the new theoretical improvements upon the results of Dasgupta et al. for explainable clustering, under the k-medians and k-means objectives (with extensions to other norms as well, and with new nearly-tight upper and lower bounds). Therefore, I recommend accepting the paper.

The main concern has to do with an overlap in results with a paper from ICML 2021 by Makarychev and Shen on the same topic. However, the prior paper did not appear publicly until well after the NeurIPS deadline, and therefore, the papers have been classified as independent work. In the next version of the paper, I strongly encourage the authors to update the related work discussing the 3-4 seemingly independent improvements to this problem, and in particular, pointing out any quantitative or qualitative similarities or differences between the works.

**Consistency Experiment:**

NeurIPS has a long history of experimentation. In 2014, NeurIPS ran an experiment in which 10% of submissions were reviewed by two independent committees to quantify the randomness in the review process. This year, we repeated a variant of this experiment to see how the quality of the review process has changed over time.  This paper was part of the experiment and was therefore assigned to two committees (consisting of reviewers, an Area Chair, and a Senior Area Chair) that reached independent decisions.  If both committees made the same recommendation, this recommendation was followed. If a single committee recommended acceptance, the paper was accepted (with the exception of a few cases in which the other committee identified what we considered a fatal flaw, e.g., an error in a key result).

This copy’s committee reached the following decision: **Accept (Spotlight)**

The other committee assigned to the paper recommended **Accept (Poster)**.  You can find the other set of reviews, along with any follow up discussion with the authors here:
https://openreview.net/forum?id=eW8HEhY9F7C